# Integrative QTL analysis of gene expression and chromatin accessibility identifies multi-tissue patterns of genetic regulation

Gregory R. Keele[1,2,3☯], Bryan C. Quach[1,2,4☯], Jennifer W. Israel[2], Grace A. Chappell[5], Lauren Lewis[5], Alexias Safi[6,7], Jeremy M. Simon[2], Paul Cotney[2], Gregory E. Crawford[6,7], William Valdar[2,8‡*], Ivan Rusyn[5‡*], Terrence S. Furey[2,8,9‡*]

**1** Curriculum in Bioinformatics and Computational Biology, University of North Carolina at Chapel Hill, Chapel Hill, North Carolina, United States of America, **2** Department of Genetics, University of North Carolina at Chapel Hill, Chapel Hill, North Carolina, United States of America, **3** The Jackson Laboratory, Bar Harbor, Maine, United States of America, **4** Center for Omics Discovery and Epidemiology, Research Triangle Institute (RTI) International, Research Triangle Park, North Carolina, United States of America, **5** Department of Veterinary Integrative Biosciences, Texas A&M University, College Station, Texas, United States of America, **6** Department of Pediatrics, Duke University, Durham, North Carolina, United States of America, **7** Center for Genomic and Computational Biology, Duke University, Durham, North Carolina, United States of America, **8** Lineberger Comprehensive Cancer Center, University of North Carolina at Chapel Hill, Chapel Hill, North Carolina, United States of America, **9** Department of Biology, University of North Carolina at Chapel Hill, Chapel Hill, North Carolina, United States of America

☯ These authors contributed equally to this work.
‡ WV, IR, and TSF also contributed equally to this work.
* william.valdar@unc.edu (WV); irusyn@tamu.edu (IR); tsfurey@email.unc.edu (TSF)

**Data Availability Statement:** The processed data necessary to generate the results reported here are available at figshare (doi:10.6084/m9.figshare.9985514). Curated results data and code to

## Abstract

Gene transcription profiles across tissues are largely defined by the activity of regulatory elements, most of which correspond to regions of accessible chromatin. Regulatory element activity is in turn modulated by genetic variation, resulting in variable transcription rates across individuals. The interplay of these factors, however, is poorly understood. Here we characterize expression and chromatin state dynamics across three tissues—liver, lung, and kidney—in 47 strains of the Collaborative Cross (CC) mouse population, examining the regulation of these dynamics by expression quantitative trait loci (eQTL) and chromatin QTL (cQTL). QTL whose allelic effects were consistent across tissues were detected for 1,101 genes and 133 chromatin regions. Also detected were eQTL and cQTL whose allelic effects differed across tissues, including local-eQTL for *Pik3c2g* detected in all three tissues but with distinct allelic effects. Leveraging overlapping measurements of gene expression and chromatin accessibility on the same mice from multiple tissues, we used mediation analysis to identify chromatin and gene expression intermediates of eQTL effects. Based on QTL and mediation analyses over multiple tissues, we propose a causal model for the distal genetic regulation of *Akr1e1*, a gene involved in glycogen metabolism, through the zinc finger transcription factor *Zfp985* and chromatin intermediates. This analysis demonstrates the complexity of transcriptional and chromatin dynamics and their regulation over multiple tissues, as well as the value of the CC and related genetic resource populations for identifying specific regulatory mechanisms within cells and tissues.

generate the results within the manuscript and supplementary tables and figures are available in the Supporting Information files. The unprocessed RNA-seq and ATAC-seq data are available from Gene Expression Omnibus (GEO) under accession number GSE140873.

**Funding:** This work was funded by grants from the National Institute of Environmental Health Sciences (NIEHS; https://www.niehs.nih.gov/index.cfm) (R01-ES023195 to IR, TSF, and GEC and P30-ES025128), with GRK and WV funded by grants from the National Institute of General Medical Sciences (NIGMS; https://www.nigms.nih.gov) (R01-GM104125 and R35-GM127000 to WV). The funders had no role in study design, data collection and analysis, decision to publish, or preparation of the manuscript.

**Competing interests:** The authors have declared that no competing interests exist.

## Author summary

Genetic variation can drive alterations in gene expression levels and chromatin accessibility, the latter of which defines gene regulatory elements genome-wide. The same genetic variants may associate with both molecular events, and these may be connected within the same causal path: a variant that reduces promoter region chromatin accessibility, potentially by affecting transcription factor binding, may lead to reduced expression of that gene. Moreover, these causal regulatory paths can differ between tissues depending on functions and cellular activity specific to each tissue. We identify cross-tissue and tissue-selective genetic regulators of gene expression and chromatin accessibility in liver, lung, and kidney tissues using a panel of genetically diverse inbred mouse strains. Further, we identify a number of candidate causal mediators of the genetic regulation of gene expression, including a zinc finger protein that helps silence the *Akr1e1* gene. Our analyses are consistent with chromatin accessibility playing a role in the regulation of transcription. Our study demonstrates the power of genetically diverse, multi-parental mouse populations, such as the Collaborative Cross, for large-scale studies of genetic drivers of gene regulation that underlie complex phenotypes, as well as identifying causal intermediates that drive variable activity of specific genes and pathways.

## Introduction

Determining the mechanisms by which genetic variants drive molecular, cellular, and physiological phenotypes has proved to be challenging [1]. These mechanisms can be informed by genome-wide experiments that provide data on variations in molecular and cellular states in genotyped individuals. Most examples of such data, though, are largely observational, due in part to constraints of specific populations (*e.g.*, humans), the limitations of existing experimental technologies, and the challenge of coordinating large numbers of experiments with multiple levels of data [2]. One approach to shed light on these dynamics is to pair complementary datasets from the same individuals and perform statistical mediation analysis (*e.g.*, [3–5]), which has increasingly been used in genomics [6]. These analyses can identify putative causal relationships rather than correlational interactions, providing meaningful and actionable targets in terms of downstream applications in areas such as medicine and agriculture.

In human data, co-occurence of QTL across various multi-omic data has been used to assess potentially related and connected biological processes; examples include gene expression with chromatin accessibility [7] or regulatory elements [8], and ribosome occupancy with protein abundances [9]. More formal integration through statistical mediation analyses has also been used to investigate relationships between levels of human biological data, such as distal genetic regulation through local gene expression [10, 11], and eQTL with regulatory elements [12–14] and physiological phenotypes, such as cardiometabolic traits [15].

Though genetic association studies of human populations have been highly successful [16], animal models allow for more deliberate control of confounding sources of variation, including experimental conditions and population structure, and as such provide a potentially powerful basis for detecting associations and even causal linkages. Towards this end, genetically-diverse mouse population resources have been established, including the Collaborative Cross (CC) [17–19] and the Diversity Outbred (DO) population [20]. The CC and the DO are multi-parental populations (MPP), derived from the same eight founder strains (short names in parentheses): A/J (AJ), C57BL/6J (B6), 129S1/SvImJ (129), NOD/ShiLtJ (NOD), NZO/H1LtJ (NZO), CAST/EiJ (CAST), PWK/PhJ (PWK), and WSB/EiJ (WSB). The CC are recombinant

inbred strains and therefore replicable across and within studies; the DO are largely heterozygous, outbred animals, bred with a random mating strategy that seeks to maximize diversity. MPPs similar to the CC or DO have also been developed in other species, including rats, *Arabidopsis*, *Drosophila*, and yeast, and the use of MPPs in model organism research has accelerated significantly in recent years ([21] and refs therein).

As with humans, it is only recently that studies on MPPs have used mediation analysis to connect genetic variants with different levels of genomic data. A genome-wide mediation approach in 192 DO mice was used to link transcriptional and post-translational regulation of protein levels [22]. CC mice were then used to confirm results by showing correspondence with estimates of founder haplotype effects from each of the related populations. More recently, mediation analysis was used to connect chromatin accessibility with gene expression in embryonic stem cells derived from DO mice [23]. The CC and DO, and MPPs more broadly, have well-characterized haplotype structures that provide a unique opportunity for studying mediation at the haplotype level. This is potentially advantageous because haplotypes can capture genetic variation and its effects more comprehensively than can individual SNPs [24], the latter being the predominant basis for comparable mediation analyses in humans.

Here we use a sample composed of a single male mouse from 47 CC strains to investigate dynamics between gene expression and chromatin accessibility, as determined by Assay for Transposase Accessible Chromatin sequencing (ATAC-seq), in lung, liver, and kidney tissues. We detect QTL underlying gene expression and chromatin accessibility variation across the strains and assess support for mediation of the effect of eQTL through chromatin accessibility using a novel implementation of previous methods used in the DO [22]. Additionally, we detect gene mediators of distal-eQTL. These findings demonstrate the experimental power of the CC resource for integrative analysis of multi-omic data to determine genetically-driven phenotype variation, despite limited sample size, and provide support for continued use of the CC in larger experiments going forward.

## Results

### Differential gene expression and chromatin accessibility

**Gene expression and chromatin accessibility cluster by tissue.** Gene expression and chromatin accessibility were measured with RNA-seq and ATAC-seq assays, respectively, from whole lung, liver, and kidney tissues in one male mouse from each of 47 CC strains (Fig 1). (The use of only male mice was due to practical constraints; results for females may differ [22].) Each tissue has a distinct function and we expected those differences to be reflected in the data. This was borne out by principal components analysis (PCA) of each of the gene expression and chromatin accessibility profiles, which showed that the samples clearly clustered by tissue (S1 Fig).

**Differentially expressed genes strongly correspond with accessible chromatin regions.** Differential expression (DE) and differentially accessible region (DAR) analysis were performed between the three tissues (S1 Table) and revealed between 3,564–5,709 DE genes and 28,048–40,797 DARs (FDR $\leq$ 0.1). For both expression and chromatin accessibility, liver and kidney tissues were the most similar, whereas lung and liver were the most distinct, also seen in the PCA plots (S1 Fig). Pathway analyses showed many between-tissue differences related to metabolic and immune-related pathways (FDR $\leq$ 0.1), reflecting the distinct demands of each tissue. Energy metabolism pathways were more active in liver and kidney and immune-related pathways were more pronounced in lung, as expected. We compared the concordance between DE genes and DARs genome-wide and observed that most DE gene promoters do not show significant differences in chromatin accessibility (S2 Fig). In cases with

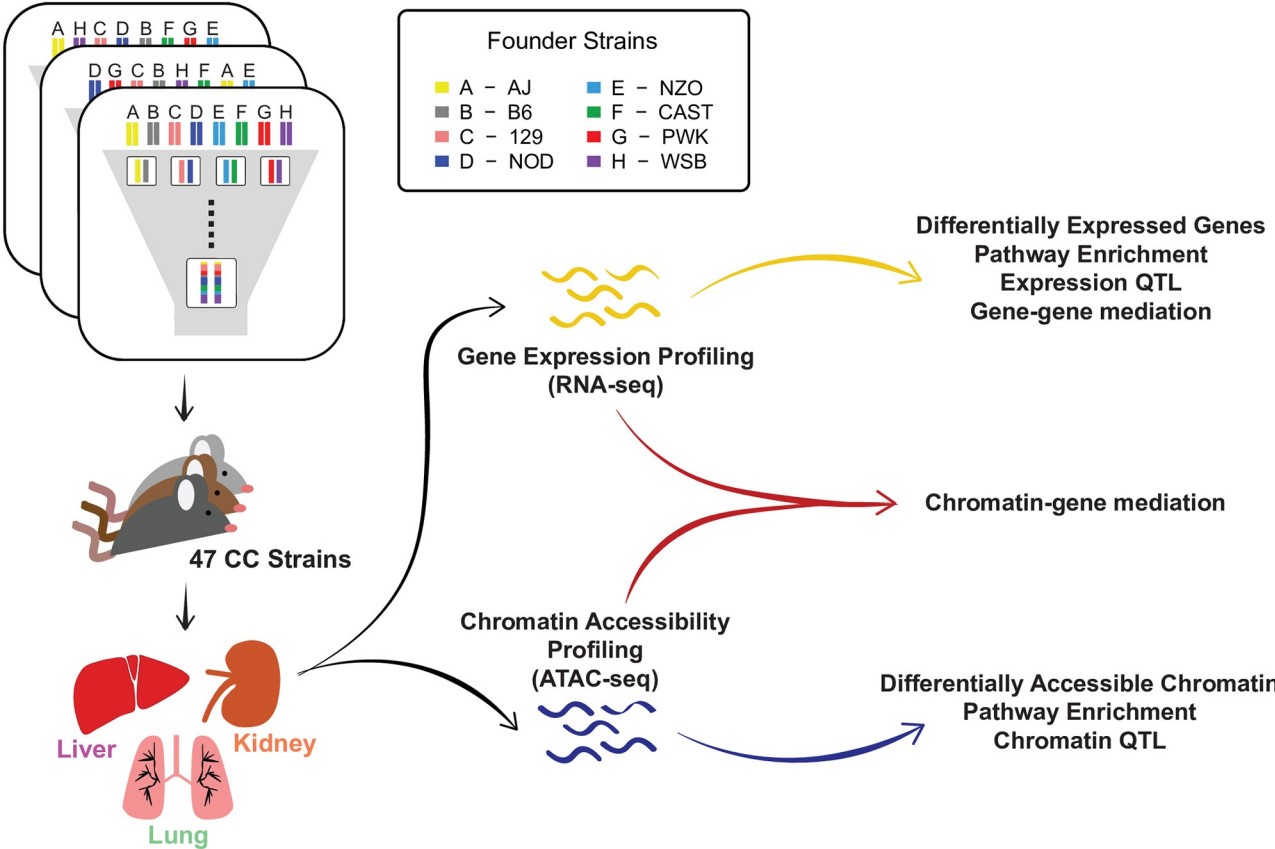

**Fig 1. Diagram of the experiment and analyses.** RNA-seq and ATAC-seq were performed using liver, lung, and kidney tissues of males from 47 CC strains. Each CC strain was derived from an inbreeding funnel, and thus represents a recombinant inbred mosaic of the initial eight founder haplotypes. Differential analyses followed by pathway enrichment analyses were performed to identify biological pathways enriched in differentially expressed genes and accessible chromatin regions. QTL and mediation analyses were performed to identify regions that causally regulate gene expression and chromatin accessibility.

significant variability in accessibility at the promoter of a DE gene, though, the vast majority agree in direction (*i.e.*, higher expression with greater accessibility).

## QTL detection

**Gene expression.** The impact of genetic variation on gene expression was evaluated by eQTL mapping. This was done at three levels of stringency and emphasis: 1) at the level of the local region of a gene, defined as within 10Mb of the gene transcription start site (TSS), and hereafter termed Analysis L; 2) at the level of the chromosome on which the gene is located (Analysis C); and 3) at the level of the genome (Analysis G) (details in Methods). After filtering out lowly expressed genes, the number of genes examined in eQTL mapping was 8401 for liver, 11357 for lung, and 10092 for kidney (UpSet plot [25] in S3A Fig).

Analysis L detected local-eQTL for 19.8% of genes tested in liver, 16.6% in lung, and 20.8% in kidney (S2 Table). Local-eQTL for most genes were observed in only one tissue (S4A Fig). Analysis C, which was more stringent, additionally detected intra-chromosomal distal-eQTL, while Analysis G, the most stringent, additionally detected inter-chromosomal distal-eQTL (S2 Table). Genomic locations of eQTL detected for each tissue, excluding the intra-chromosomal distal-eQTL detected by Analysis C, are shown in Fig 2A[top]. See S5 Fig and S3 Table for eQTL counts with FDR ≤ 0.2.

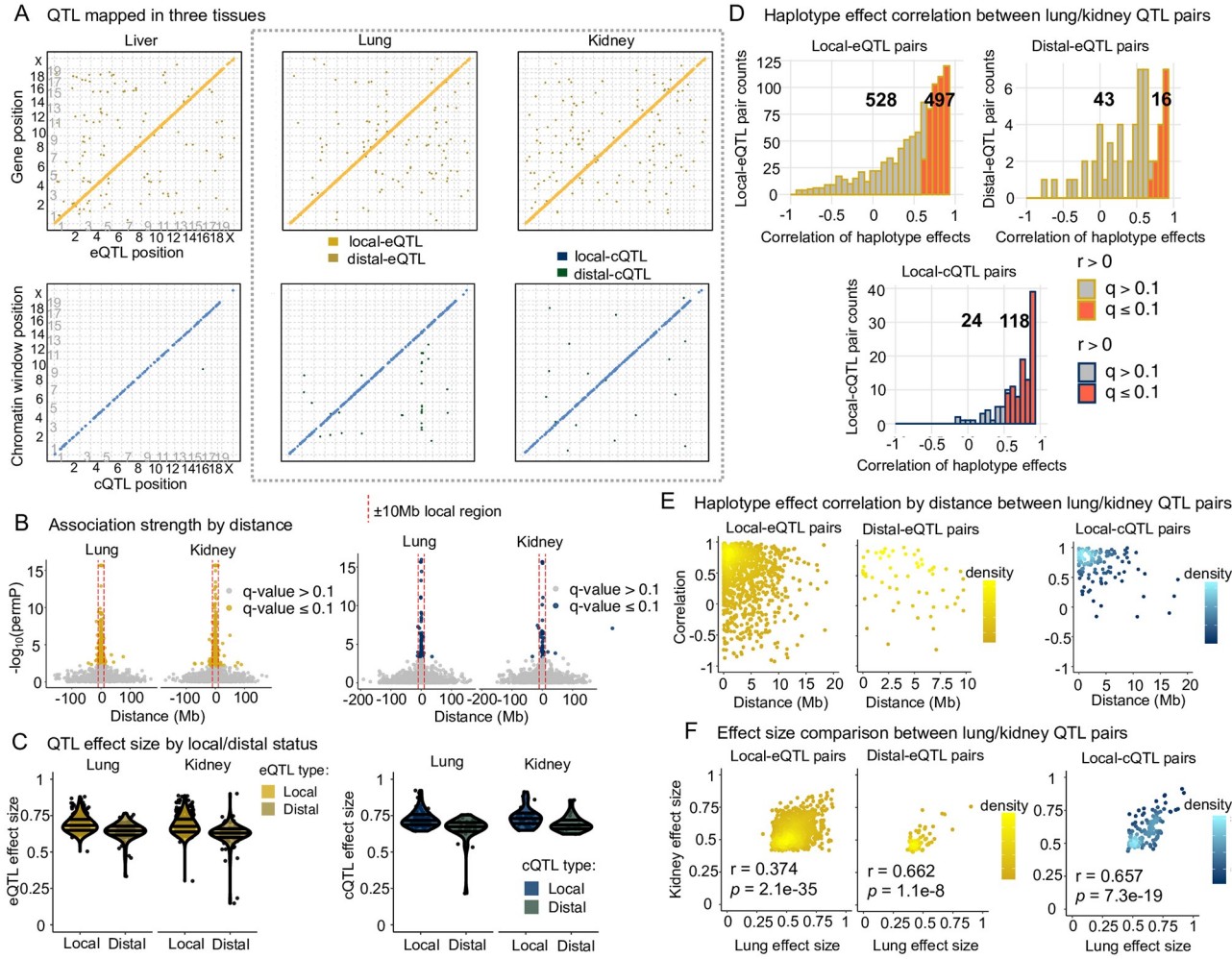

**Fig 2. QTL mapping results and comparison of matched pairs from lung and kidney.** (A) Detected QTL are largely local for both gene expression and chromatin accessibility. QTL detected through all analyses, excluding intra-chromosomal distal-QTL detected through Analysis C, are included. The y-axis represents the genomic position of the gene or chromatin site, and the x-axis represents the genomic position of the QTL. Local-QTL appear as dots along the diagonal and distal-QTL as off-diagonal dots. The gray dashed box highlights lung and kidney QTL results, which are further explored in B-F. (B) Highly significant QTL map nearby the gene TSSs and chromatin window midpoints. Results from Analysis G are shown here. The red dashed line represents ± 10Mb from the trait genomic coordinate used to classify a QTL as local or distal. See S6 Fig for all tissues as well as results from Analysis C. (C) Local QTL have larger effect sizes than distal. Only results from Analysis G are included here. See S7 Fig for all tissues and results from Analysis C. (D) Consistent genetic regulation of gene expression and chromatin accessibility was observed between lung and kidney tissue, based on an enrichment in positively correlated founder haplotype effects for QTL paired across the two tissues. QTL from all the Analyses were considered here. See S10 Fig for all tissue pairings. (E) QTL observed in both lung and kidney that are highly correlated tend to map close to each other, consistent with representing genetic variation active in both tissues. See S11 Fig for all tissue pairings. (F) The effect sizes of QTL paired from lung and kidney are significantly correlated. See S9 Fig for all tissue comparisons.

**Chromatin accessibility.** To determine genetic effects on chromatin structure, genomic regions were divided into ∼300 base pair windows and analyses similar to eQTL mapping were used to detect chromatin accessibility QTL (cQTL). After filtering regions with low signal across most samples, the number of chromatin regions tested were 11448 in liver, 24426 in lung, and 17918 in kidney. The overlap in chromatin windows tested across tissues is described in S3B Fig. The differences in genes and chromatin regions tested within each tissue likely reflects both biological and technical factors that distinguish the tissue samples. Overall, there were substantially fewer cQTL detected compared with eQTL for all tissues (Fig 2A[bottom];

S4 and S5 Tables). As with eQTL, cQTL were more likely to be local than distal (66—94.1% local-cQTL for Analysis G; 75—90% local-cQTL for Analysis C).

**Local-QTL have stronger effects than distal-QTL.** For QTL detected on the same chromosome as the gene or chromatin region (intra-chromosomal), the strongest associations were observed within 10Mb of the gene TSS or chromatin window midpoint (Fig 2B and S6 Fig). Intra-chromosomal distal-QTL had reduced statistical significance, more consistent with inter-chromosomal distal-QTL. This dynamic between local- and distal-QTL is also observed when using the QTL effect size as a measure of strength, which is likely biased upward due to the Beavis effect in 47 strains [26], shown in Fig 2C and S7 Fig.

**QTL driven by extreme effects of CAST and PWK.** For all QTL we estimated the effects of the underlying (founder) haplotypes. Consistent with previous studies (*e.g.*, [27]), the CAST and PWK haplotypes had higher magnitude effects compared with the classical inbred strains. This pattern was observed for both local- and distal-eQTL (S8A Fig) and local-cQTL; the numbers of detected distal-cQTL were too low to produce clear trends (S8B Fig).

## QTL paired across tissues and correlated haplotype effects

For a given trait, QTL from different tissues were paired if they co-localized to approximately the same genomic region. In the case of local-QTL, both had to be within the local window, defined as 10Mb up or downstream of the gene TSS or chromatin region midpoint; thus, the maximum possible distance between them was 20Mb. In the case of distal-eQTL, they had to be within 10Mb of each other. Consistency between these cross-tissue QTL pairs was assessed by calculating the correlation between their haplotype effects (FDR ≤ 0.1; Fig 2D and S10 Fig), with significant positive correlations implying that the paired QTL acted similarly and likely represent multi-tissue QTL. For local-eQTL, we found significant haplotype effect correlations between 346 of 761 possible pairs (45.5%) in liver/lung, 623 of 1206 (51.7%) in liver/kidney, and 497 of 1025 (48.5%) in lung/kidney. For distal-eQTL, we found significant correlations between 21 of 61 possible pairs (31.8%) in liver/lung, 34 of 120 (28.8%) in liver/kidney, and 16 of 59 (27.1%) in lung/kidney. For cQTL, the vast majority of correlated pairs were local, with 47 of 55 (85.5%) possible pairs in liver/lung, 48 of 56 (85.7%) in liver/kidney, and 118 of 142 (83.1%) in lung/kidney being significantly correlated. Only 4 distal-cQTL pairs were observed, all between lung and kidney, with three of the four pairs possessing nominal correlation $p$-values ≤ 0.05. The effect sizes of paired QTL varied across tissues, though for any given tissue pairing, the effect sizes were significantly correlated (least significant $p$-value = $4.6 \times 10^{-8}$; Fig 2F and S9 Fig). No significantly negatively correlated QTL pairs were detected after accounting for multiple testing (FDR ≤ 0.1).

***Cox7c*: Consistent haplotype effects for multi-tissue local-eQTL.** Cytochrome c oxidase subunit 7C (*Cox7c*) is an example of a gene that possessed local-eQTL with highly correlated effects in all three tissues (Fig 3A). The local-eQTL consistently drove higher expression when the CAST haplotype was present, intermediate expression with 129, NOD, and NZO haplotypes, and lower expression with AJ, B6, PWK, and WSB haplotypes. Though the haplotype effects on relative expression levels within each tissue were consistent, we noted that the expression level was significantly higher in liver compared with both lung ($q = 5.41 \times 10^{-18}$) and kidney ($q = 6.25 \times 10^{-19}$). Ubiquitin C (*Ubc*) is another example of a gene with consistent local-eQTL detected in all three tissues (S12 Fig).

***Slc44a3* and *Pik3c2g*: Tissue-specific haplotype effects for local-eQTL.** We found instances where haplotype effects across the three tissues were inconsistent, referring to these as tissue-specific effects (within this subset of tissues). The strongest support of tissue-specificity would be given by QTL pairs whose effects are uncorrelated. For example, the solute carrier

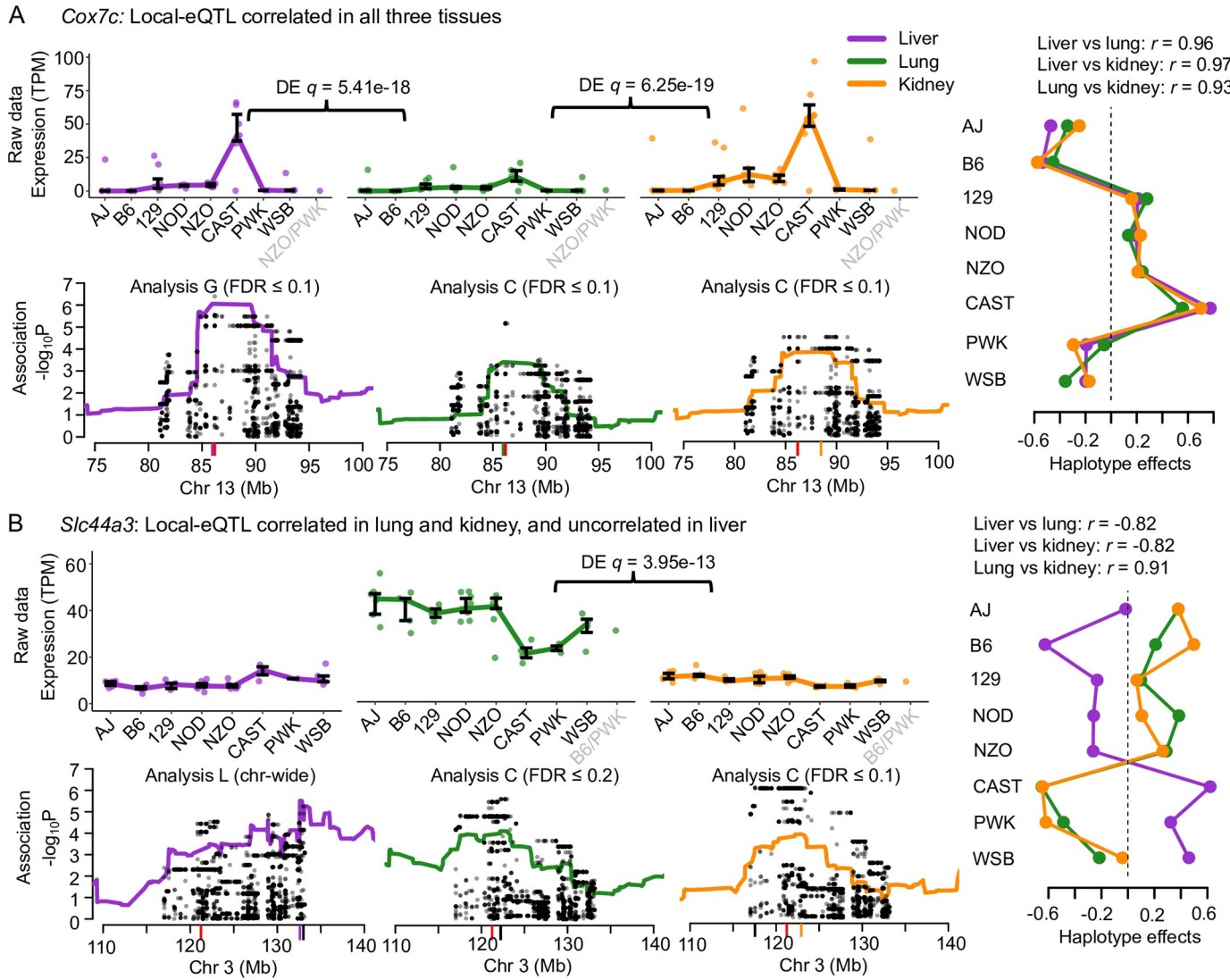

**Fig 3. Examples of genes with local-eQTL observed in all three tissues.** (A) *Cox7c* possesses local-eQTL with highly correlated haplotype effects across all three tissues, supportive of shared causal origin. (B) *Slc44a3* has a more complicated pattern of local-eQTL haplotype effects across the tissues, with correlated effects shared between lung and kidney, and transgressive effects in the liver eQTL by comparison, consistent with distinct causal variants comparing liver to lung and kidney. For both genes, the expression data are plotted with bars representing the interquartile ranges of the most likely founder haplotype pair (diplotype). Differential expression between tissues is highlighted. The haplotype association for each tissue is also included near the gene TSS with variant association overlaid. The most statistically rigorous method that detected the QTL (Analysis L, C, or G) is also included. The red tick represents the gene TSS, the black tick represents the variant association peak, and the colored tick represents the haplotype association peak. Haplotype effects, estimated as constrained best linear unbiased predictions (BLUPs).

family 44, member 3 (*Slc44a3*) gene has local-eQTL effects that are correlated in lung and kidney but anticorrelated in liver (Fig 3B), suggesting the liver eQTL could be transgressive [28] relative to the eQTL in lung and kidney, whereby the effects of the haplotypes are reversed. For *Slc44a3*, CAST, PWK, and WSB haplotypes result in higher expression in liver but lower expression in lung and kidney. The local-eQTL for *Slc44a3* were more similar in location in lung and kidney whereas the liver eQTL was more distal to the gene TSS. Overall, the expression data, estimated effects, and patterns of association are consistent with lung and kidney sharing a causal local-eQTL that is distinct from the one in liver.

The CC founder strains all possess contributions from three mouse subspecies of *M. musculus*: *domesticus* (*dom*), *castaneus*, (*cast*), and *musculus* (*mus*) [29]. Allele-specific gene

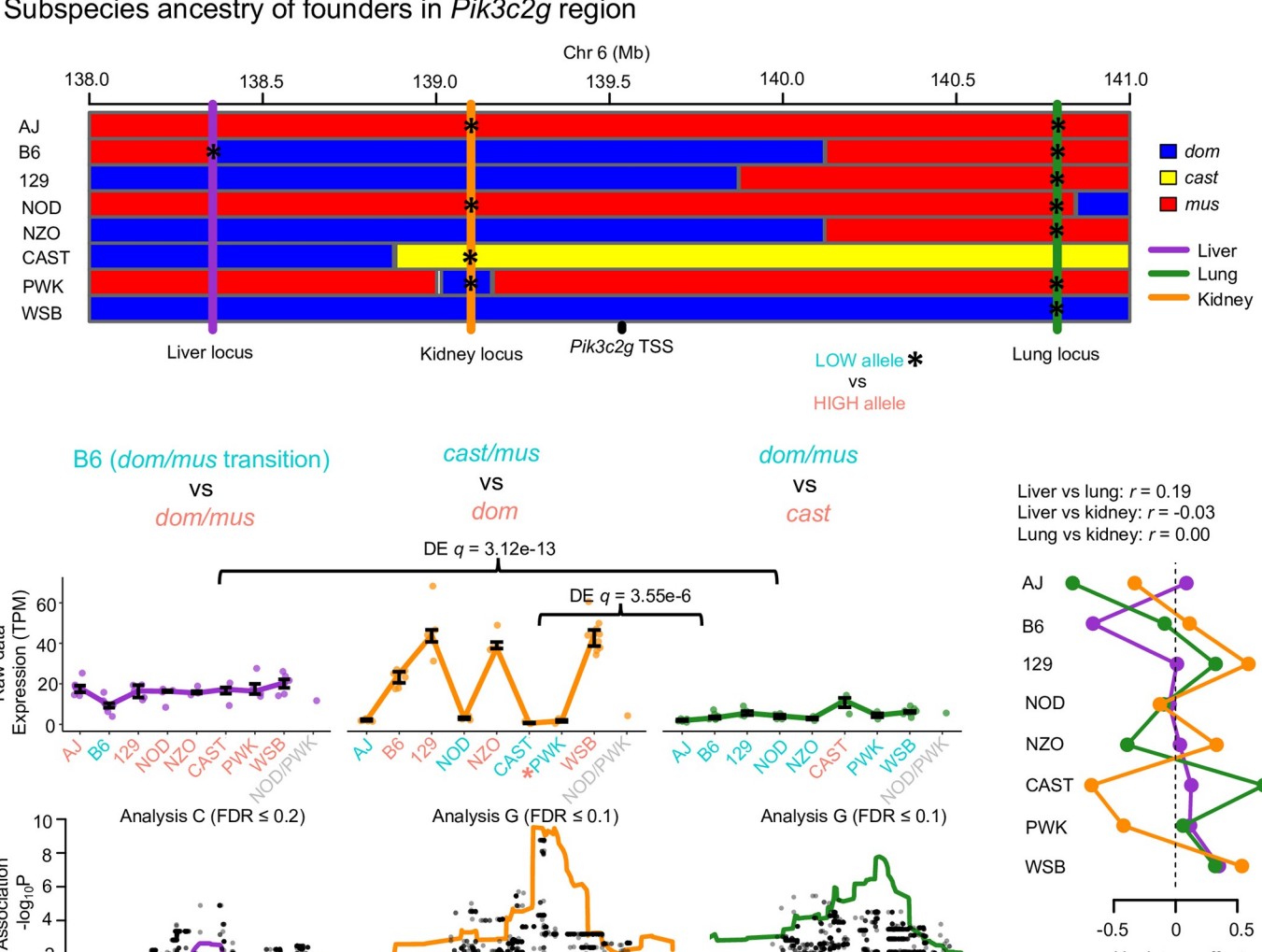

**Fig 4. *Pik3c2g* possesses tissue-specific local-eQTL.** Local-eQTL for *Pik3c2g* were detected in all three tissues in the 3Mb region surrounding its TSS. The genomes of the CC founders can be simplified in terms of contributions from three subspecies lineages of *M. musculus*: *dom* (blue), *cast* (yellow), and *mus* (red). The effects of each local-eQTL matched the subspecies contributions near the eQTL coordinates, with low expressing subspecies alleles colored teal and high expressing alleles colored salmon, consistent with local-eQTL for *Pik3c2g* being distinct and tissue-specific. The gene expression data are represented as interquartile range bars, categorized based on most likely diplotype at the eQTL for each CC strain. Haplotype and variants associations are included for each tissue, with the red tick representing the *Pik3c2g* TSS, black ticks representing variant association peak, and colored ticks representing the haplotype association peak. The most rigorous procedure to detect each QTL (Analysis L, C, or G) is reported. Haplotype effects, estimated as constrained BLUPs, were consistent with the expression data, and uncorrelated across the tissues.

expression in mice descended from the CC founders often follow patterns that matched the subspecies inheritance at the gene regions [30, 31]. We found that phosphatidylinositol-4-phosphate 3-kinase catalytic subunit type 2 gamma (*Pik3c2g*), a gene of interest for diabetes-related traits [32], had tissue-specific local-eQTL in all three tissues that closely matched the subspecies inheritance at their specific genomic coordinates. The local-eQTL in all three tissues were all pair-wise uncorrelated (Fig 4). Further, expression of *Pik3c2g* varied at statistically significant levels for liver versus lung ($q = 3.12 \times 10^{-13}$) and lung versus kidney ($q = 3.55 \times 10^{-6}$). In lung, the CAST haplotype (*cast* subspecies lineage) resulted in higher expression, consistent with a *cast* versus *dom*/*mus* allelic series. In kidney, the B6, 129, NZO,

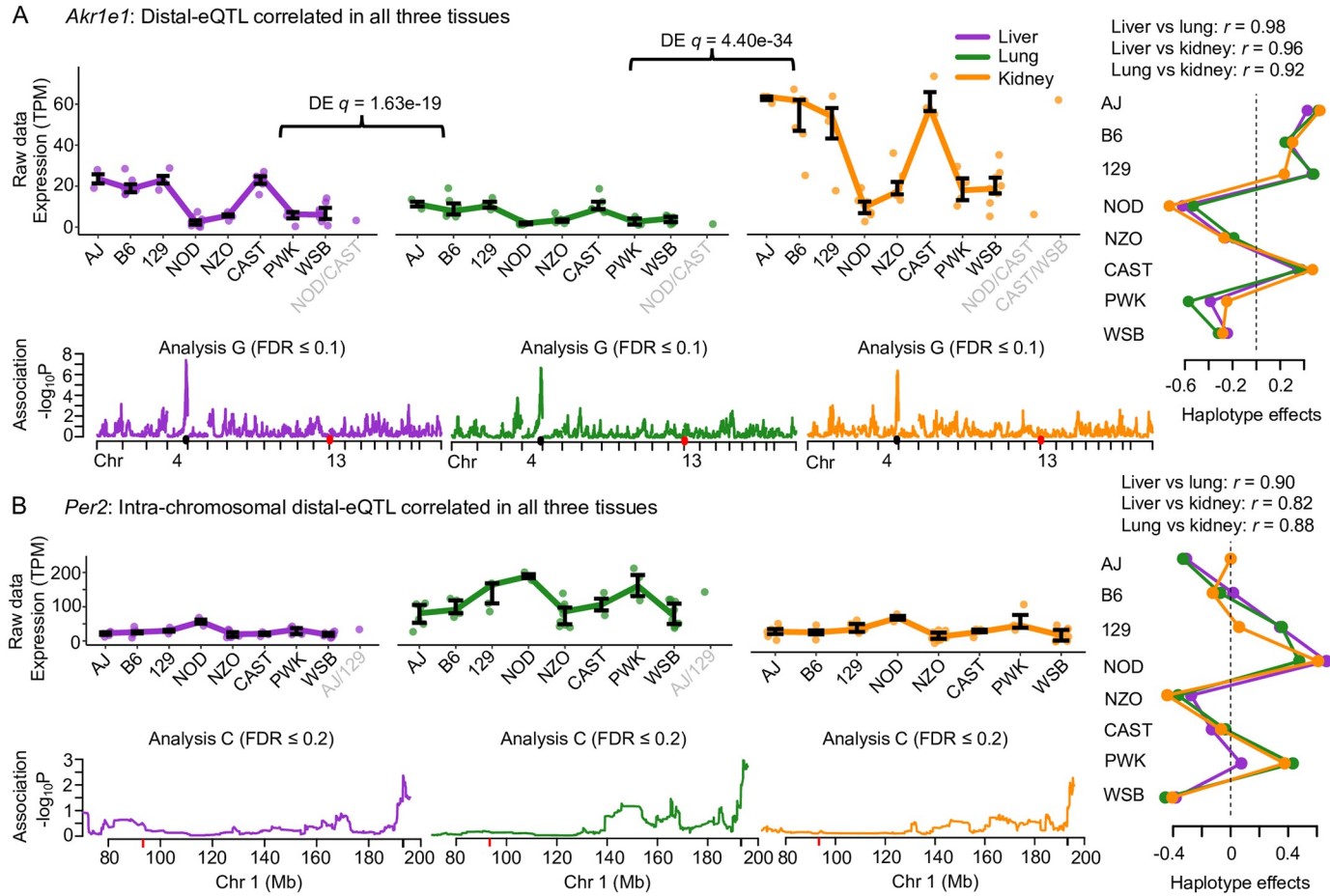

**Fig 5. Examples of genes with distal-eQTL effect patterns across tissues.** (A) *Akr1e1* has highly significant distal-eQTL detected on chromosome 4 in all three tissues with correlated haplotype effects. (B) *Per2* has intra-chromosomal distal-eQTL leniently detected 100Mb away from the TSS, also with highly correlated haplotype effects across the tissues, providing further support that the distal-eQTL are not false positives. Expression data are represented as interquartile ranges for most likely diplotype, with differential expression noted when significant. Haplotype associations for each tissue distal-eQTL combination are shown, with the most rigorous statistical procedure for detection reported (Analysis L, C, or G). Red ticks signify the gene TSS and black ticks represent that eQTL peak. Fit haplotype effects, estimated as BLUPs, are included, along with the pairwise correlations of the eQTL.

and WSB haplotypes (*dom* subspecies) resulted in higher expression, whereas AJ and NOD (*mus*), CAST (*cast*), and PWK (*dom*) haplotypes showed almost no expression, mostly consistent with a *dom* versus *cast*/*mus* allelic series. The PWK founder appears inconsistent, but we note that the QTL peak was in a small *dom* haplotype block interspersed within a broader *mus* region. The expression level in kidney of *Pik3c2g* in PWK was low, similar to AJ and NOD, suggesting that the causal variant may be located in the nearby region, where all three have *mus* inheritance and, notably, where the peak variant association occurred. In liver, the B6 haplotype resulted in lower expression compared with the other haplotypes. Interestingly, the liver eQTL was in a region that contained a recombination event between *dom* and *mus*, only present in B6, which may explain the unique expression pattern.

**_Akr1e1_ and _Per2_: Consistent haplotype effects for multi-tissue distal-eQTL.** Haplotype effects that correlate across tissues can provide additional evidence for distal-QTL, even those with marginal significance in any single tissue. For example, the aldo-keto reductase family 1, member E1 (*Akr1e1*; Fig 5A) gene is located on chromosome 13 with no local-eQTL detected in any tissue. Distal-eQTL in all tissues were detected for *Akr1e1* that localized to the same region on chromosome 4. The haplotype effects of the distal-eQTL were all highly correlated,

with AJ, B6, 129, and CAST haplotypes corresponding to higher *Akr1e1* expression. Across tissues, expression was significantly higher in liver and kidney compared with lung ($q = 1.63 \times 10^{-19}$ and $q = 4.40 \times 10^{-34}$, respectively). Another example is the Period circadian clock 2 (*Per2*; Fig 5B) gene, which possessed intra-chromosomal distal-eQTL approximately 100Mb away from the TSS that were detected in all three tissues at a lenient chromosome-wide significance (Analysis C; FDR ≤ 0.2). The haplotype effects were significantly correlated among the tissues, characterized by high expression with the NOD and PWK haplotypes present. Together, these findings provide strong validation for the distal-eQTL, which would commonly not be detected based on tissue-stratified analyses. Other unique patterns of distal genetic regulation were detected, such as for ring finger protein 13 gene (*Rnf13*) with distal-eQTL that varied across tissues, described in S13 Fig.

### Mediation of eQTL by chromatin

An advantage of measuring gene expression and chromatin accessibility in the same mice and tissues is the subsequent ability to examine the relationships between genotype, chromatin accessibility, and gene expression using integrative methods such as mediation analysis. We considered two possible models for the mediation of eQTL effects and assessed the evidence for each in our data. In the first model, proximal chromatin state acts as a mediator of local-eQTL (Fig 6A). That is, the local-eQTL for a gene affects that gene's expression by, at least in

## A  Mediation of local-eQTL through chromatin

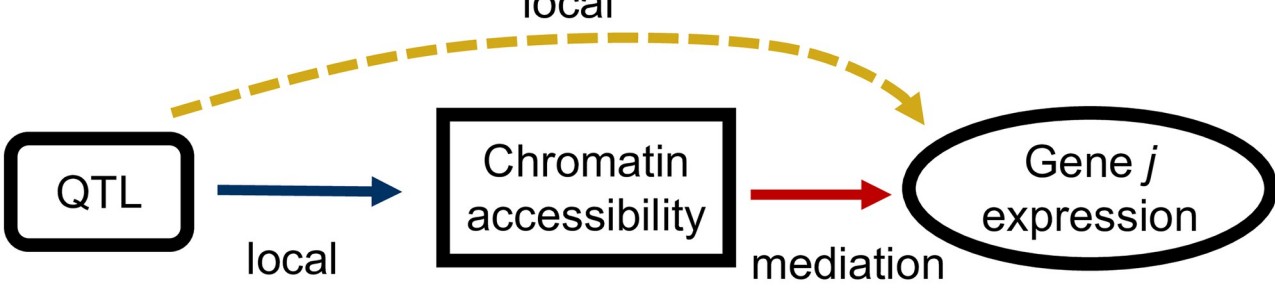

## B  Mediation of distal-eQTL through gene expression

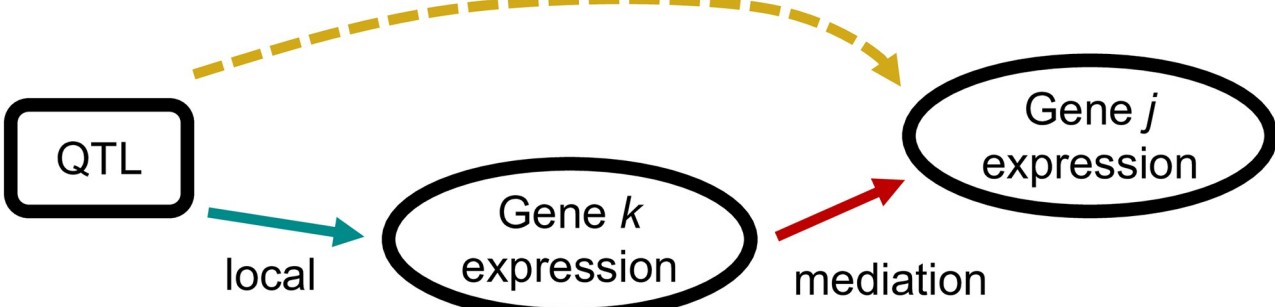

**Fig 6. Simple mediation models for the genetic regulation of gene expression.** (A) Mediation of the local-eQTL through chromatin accessibility in the region of the gene is consistent with genetic variation influencing the accessibility of gene *j* to the transcriptional machinery. (B) Mediation of distal-eQTL through the transcription of genes local to the QTL could be explained by transcription factor activity.

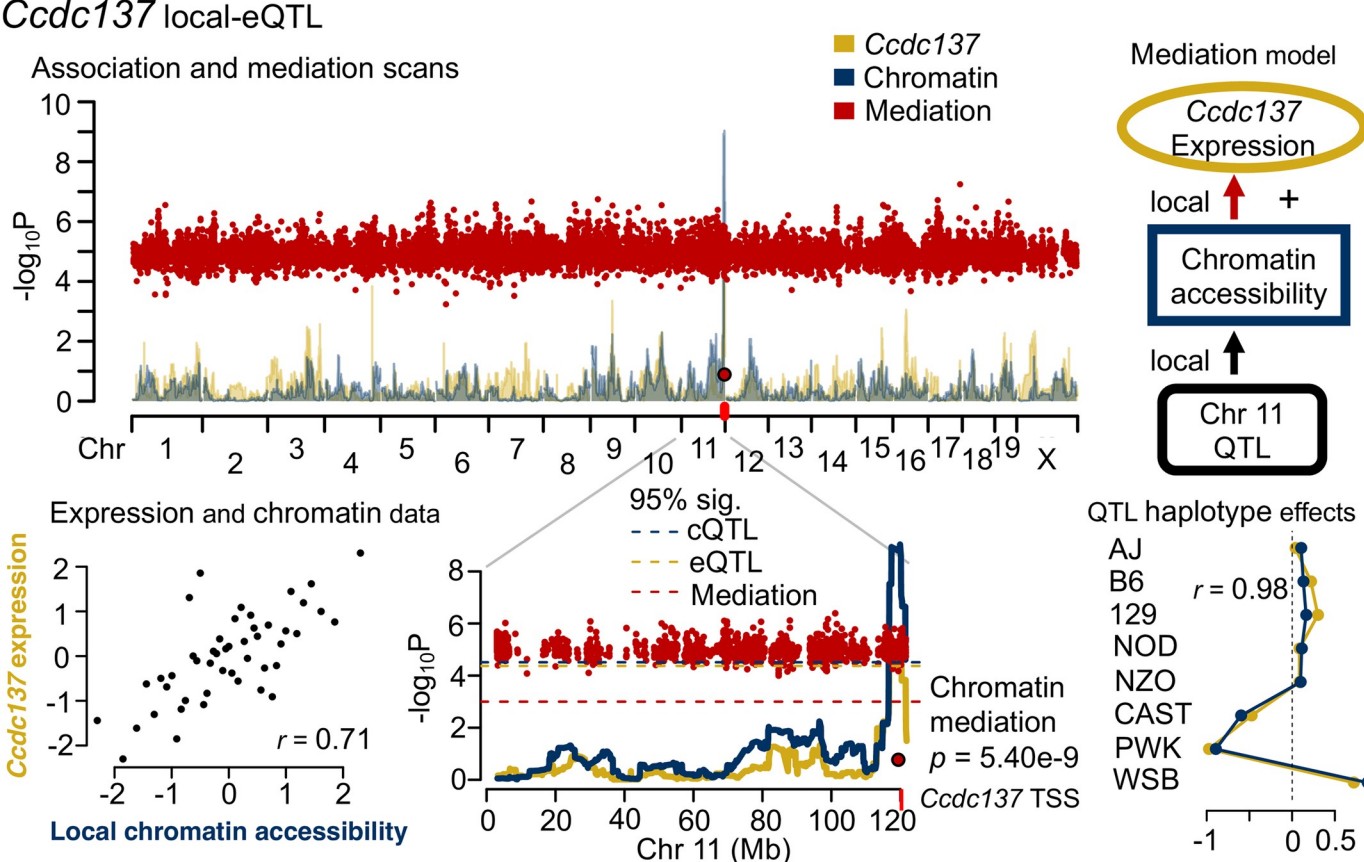

**Fig 7. *Ccdc137* local-eQTL is mediated by proximal chromatin accessibility.** *Ccdc137* expression and the chromatin accessibility in the proximal region were highly correlated ($r = 0.71$). Genome scans for *Ccdc137* expression (yellow), nearby chromatin accessibility (blue), and chromatin mediation of the *Ccdc137* local-eQTL (red) in lung tissue. The local-eQTL and local-cQTL for the chromatin region at the TSS of *Ccdc137* (red tick) are over-lapping, and have highly correlated haplotype effects ($r = 0.98$). The steep drop in the statistical association with expression, represented as logP, at the chromatin site in the mediation scan supports chromatin mediation of *Ccdc137* expression, depicted as a simple graph [top right]. The QTL and mediation signals were detected at genome-wide significance.

part, altering local chromatin accessibility. To test for this, we used an approach adapted from studies in the DO [22] and applied it to local-eQTL detected through Analysis L (see S3 Appendix for greater detail).

***Ccdc137* and *Hdhd3*: Local-eQTL driven by local chromatin accessibility.** Across the three tissues, between 13-42 local-eQTL showed evidence of mediation through proximal accessible chromatin regions at genome-wide significance, and 35-106 at chromosome-wide significance (S6 Table). The coiled-coil domain containing 137 gene (*Ccdc137*) is a strong example of this type of mediation. Fig 7 shows the genome scans that identify both the local-eQTL for *Ccdc137* and the cQTL near it. The significance of the eQTL was sharply reduced when we conditioned on chromatin accessibility, and the significance of the cQTL was stronger than the eQTL, as expected by the proposed mediator model.

Establishing mediation requires more than mere co-localization of eQTL and cQTL. For example, local-eQTL and co-localizing cQTL were identified for both haloacid dehalogenase-like hydrolase domain containing 3 (*Hdhd3*) in liver (S14A Fig) and acyl-Coenzyme A binding domain containing 4 (*Acbd4*) in kidney (S14B Fig). Comparing the statistical associations at the loci for eQTL and cQTL, and the corresponding haplotype effects, however, showed better correspondence in the case of *Hdhd3* than for *Acbd4*: a strong mediation signal was detected for *Hdhd3*, indicated by the decreased eQTL association when conditioning on the chromatin

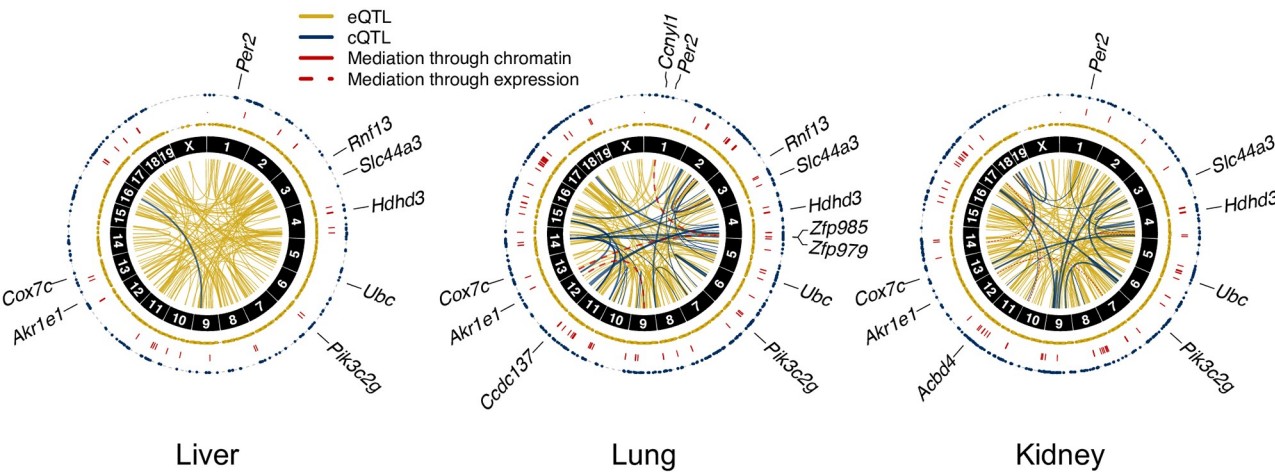

**Fig 8. Summaries of QTL and mediation analyses.** Circos plots of eQTL (yellow), cQTL (blue), and mediation (red) in lung, liver, and kidney. The two outer rings of dots represent local-eQTL and local-cQTL detected by Analysis L at chromosome-wide significance, with red lines between connecting genes and chromatin sites for which chromatin mediation was detected. The inner circle contains connections representing distal-eQTL, distal-cQTL, and gene-gene mediation from Analysis G. Thick lines represent QTL and mediators with permutation-based $p$-value (permP) $< 1 \times 10^{-5}$. The detected signals were primarily local, which also tended to be stronger than the observed distal signals. Fewer QTL and mediators were detected in liver tissue. Genes highlighted within the text, such as *Akr1e1*, are indicated at their genomic coordinates.

state, whereas mediation was not detected for *Abcd4*, consistent with expression and chromatin accessibility having causally different origins.

## Mediation of eQTL by expression

Mediation was also tested for distal-eQTL, detected through Analysis G, by evaluating the expression of nearby genes as candidate mediators [33], as shown in Fig 6B.

**Zinc finger protein intermediates detected for *Ccnyl1*.**  Eight genes were identified with mediated distal-eQTL (S7 Table). For example, the distal-eQTL of cyclin Y-like 1 (*Ccnyl1*) was mediated by the expression of zinc finger protein 979 (*Zfp979*), a putative transcription factor (S15 Fig). The haplotype effects at the distal-eQTL of *Ccnyl1* were highly correlated with those at the local-eQTL of *Zfp979*, though with a reduction in overall strength, in accordance with the proposed mediation model.

## Mediation of eQTL by both expression and chromatin

The QTL and mediation results for all three tissues are depicted as circos plots [34] in Fig 8. The local-QTL and chromatin mediators were distributed across the genome unevenly, aggregating in pockets. In particular, a high concentration of cQTL and chromatin mediation were detected in all three tissues along chromosome 17, which corresponds to the immune-related major histocompatibility (MHC) region in mouse. Most of the chromatin-mediated genes in this region, however, are not histocompatibility genes (S7 File). Patterns of distal-QTL and gene mediation vary across the three tissues, though consistent regions are also observed, such as the *Idd9* region on chromosome 4 described in [35], which contains multiple zinc finger proteins (ZFPs) and regulates genes such as *Ccnyl1* and *Akr1e1*, described in greater detail below.

**Genetic regulation of *Akr1e1* expression by a zinc finger protein and chromatin intermediates.**  As detailed above, *Akr1e1* possessed a strong distal-eQTL in all three tissues located on chromosome 4 in the region of 142.5—148.6Mb with significantly correlated haplotype effects. Mediation analysis suggested this effect is mediated in lung through activity of

Zinc finger protein 985 (ZFP985), whose gene is also located on chromosome 4 at 147.6Mb (S16 Fig).

ZFP985 possesses an N-terminal Krüppel-associated box (KRAB) domain, representing a well-characterized class of transcriptional regulators in vertebrates [36] that recruit histone deacetylases and methyltransferases, inducing a chromatin state associated with regulatory silencing. *Akr1e1* and *Zfp985* expression were negatively correlated ($r = -0.69$), consistent with ZFP985 inhibiting *Akr1e1* expression. This same distal-eQTL and mediator relationship for *Akr1e1* was observed in kidney tissue from 193 DO mice (S17 Fig). It was previously postulated that *Akr1e1* is distally regulated by the reduced expression 2 (*Rex2*) gene, also a ZFP that contains a KRAB domain and resides in the same region of chromosome 4 [37]. Our distal-eQTL overlap their *Idd9.2* regulatory region, defined as spanning 145.5—148.57Mb, mapped with NOD mice congenic with C57BL/10 (B10) [35]. The B10 haplotype was found to be protective against development of diabetes, characteristic of NOD mice. AKR1e1 is involved in glycogen metabolism and the *Idd9* region harbors immune-related genes, suggesting genes regulated by elements in the *Idd9* region may be diabetes-related. Consistent with these studies, CC strains with NOD inheritance at the distal-eQTL had low expression of *Akr1e1*, observed in all tissues. Genetic variation from B10 is not present in the CC, but the closely related B6 founder had high expression of *Akr1e1* like B10. NZO, PWK, and WSB haplotypes resulted in low *Akr1e1* expression, similar to NOD, while AJ, 129, and CAST haplotypes joined B6 as driving high expression (Fig 9). Variable expression of *Zfp985* is a strong candidate for driving these effects on *Akr1e1*.

Additional sources of genetic regulation of *Akr1e1* were observed for kidney, where a strong distal-cQTL for an accessible chromatin site corresponding to the promoter of *Akr1e1* was detected in the *Idd9.2* region on chromosome 4. Mediation analysis strongly supported chromatin accessibility in this region mediating the distal-eQTL (mediation $p = 2.18 \times 10^{-13}$; see Methods). The haplotype effects for the distal-cQTL were highly correlated with the distal-eQTL effects ($r = 0.92$). The relative magnitudes of the QTL effect sizes and mediation $p$-values (S18 Fig) support a causal model whereby increased *Zfp985* expression reduces expression of *Akr1e1* by altering chromatin accessibility near the *Akr1e1* promoter (Fig 9).

## Discussion

In this study we performed QTL and mediation analyses of gene expression and chromatin accessibility data in liver, lung, and kidney tissue samples from 47 strains of the CC. We examined correlations between haplotype effects of co-localizing QTL to identify QTL that are likely functionally active in multiple tissues as well as QTL with distinct activity across the three tissues, as is the case with the *Pik3c2g* gene, potentially representing differing active genetic variants in the local region. We detected extensive evidence of chromatin mediation of local-eQTL as well as gene expression mediators underlying distal-eQTL. One unique example is the elucidation of the genetic regulation of *Akr1e1* expression, a gene that plays a role in glycogen metabolism, involving inhibition by expression of a distal zinc finger protein mediator that contributes to reduced chromatin accessibility at the promoter of *Akr1e1*. These findings highlight the ability of integrative QTL approaches such as mediation analysis to identify interesting biological findings, including the ability to identify functional candidates for further downstream analysis.

### Fewer detected cQTL than eQTL

We found that the effect sizes of cQTL are on average lower than eQTL (see S19 Fig), as has been previously reported [22]. The reduced number of cQTL compared with eQTL is likely due to both technical and biological reasons. Technically, the RNA-seq assay measures a

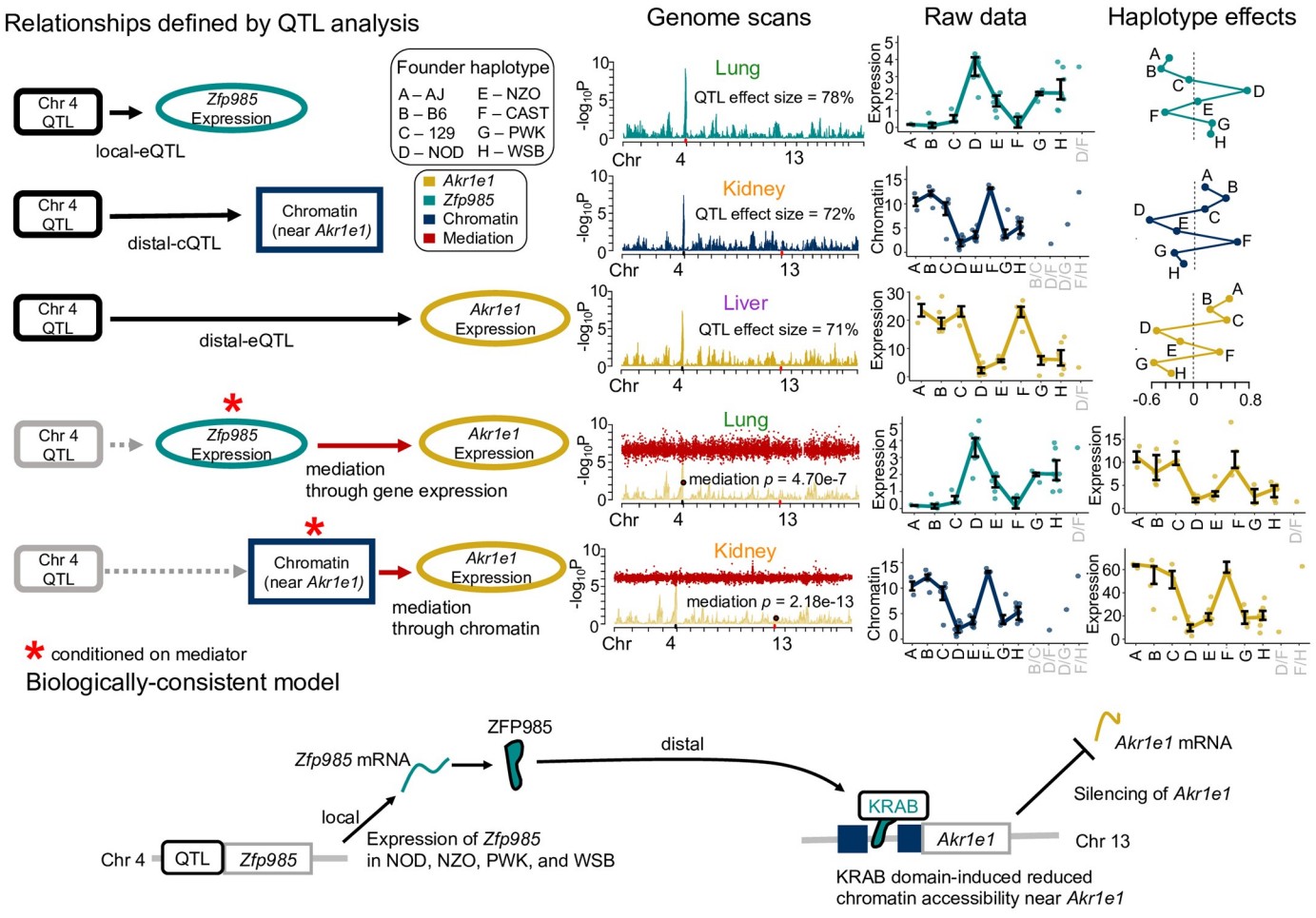

**Fig 9. Mediation model for *Akr1e1* distal-eQTL.** The genetic regulation of *Akr1e1* expression is reconstructed based on relationships observed across the three tissues. Distal-eQTL were detected in all tissues at similar levels of significance. A local-eQTL for *Zfp985* that is proximal to the *Akr1e1* distal-eQTL was observed in lung, and *Zfp985* expression was detected as an anti-correlated mediator of the distal-eQTL, consistent with ZFP985 suppressing *Akr1e1* expression. The chromatin site proximal to the *Akr1e1* TSS has a distal-cQTL detected in kidney. Chromatin accessibility at the site was found to be a significant mediator of *Akr1e1* expression. Combining associations across tissues supports a biological model whereby ZFP985, whose gene is expressed in mice with NOD, NZO, PWK, and WSB haplotypes, silences *Akr1e1* through KRAB domain-induced chromatin remodeling. QTL and mediation genome scans are included, along with sequence phenotypes as interquartile ranges categorized according to most likely diplotype, and modeled haplotype effects fit as BLUPs. The relative magnitudes of the QTL effect sizes and mediation scores are consistent with the proposed model, with *Zfp985* local-eQTL > distal-cQTL > *Akr1e1* distal-eQTL, and chromatin mediation > mediation through *Zfp985* expression.

distinct class of molecules (mRNAs) that can be accurately extracted from cells with the resulting sequence reads mapped to the transcriptome, which encompasses < 5% of the genome. In contrast, the transposon incorporation event central to ATAC-seq is enriched in, but not solely limited to, accessible chromatin. Unlike the transcriptome, accessible chromatin can occur anywhere in the genome and regions are defined empirically by the data. Thus, the signal from the assay is more variable and noisy across samples, which impedes our ability to detect cQTL. Biologically, if a variant affects the activity of a regulatory element that alters expression levels, it is expected that the mRNA levels in the sample will reflect this. By contrast, even if a variant affects chromatin accessibility, it may do so in a manner that is difficult to detect above the background noise of the assay. In recent studies of gene expression and chromatin accessibility in the adult brain from the same individuals, 2,154,331 *cis*-eQTL were found for 467 individuals [38], whereas only 6,200 cQTL were detected, albeit for only 272 individuals [39]. While the reduced number of individuals undoubtedly contributed to

this lower number of detected cQTL, it is likely that significantly fewer cQTL would be found in the same number of individuals.

## Reduced cQTL and mediation signal in liver compared with lung and kidney

Detecting QTL and mediation depends not only on sample size but also on biological and technical factors that are difficult to quantify across the tissues. Although it is possible that liver has fewer actual cQTL (and thus fewer chromatin mediators) than lung and kidney, it is also possible that the signal quality of the ATAC-seq is lower for this tissue, resulting in fewer detections due to increased technical noise. True biological differences in the number of cQTL and mediator usage among the tissues would likely reflect multi-level regulatory programs specific to each tissue, a complex subject requiring more targeted experiments than used here.

## Joint QTL analysis in multiple tissues

Multi-tissue QTL analyses are increasingly used in both humans, such as within the GTEx project (*e.g.*, [40, 41]), and in mice (*e.g.*, [42, 43]). We believe our use of formal statistical tests of the correlation coefficient between the haplotype effects of overlapping QTL is novel in defining co-localizing QTL across tissues. This method allows us to identify loci likely representing tissue-specific QTL with unique haplotype effects patterns, as demonstrated for the gene *Pik3c2g* (Fig 4). This approach can also detect consistent haplotype effects, as with the gene *Per2* (Fig 5B), which possessed only marginally significant distal-eQTL in the three tissues, suggesting jointly mapping QTL across all tissues increases power. Formal joint analysis approaches have been proposed, largely implemented for detecting SNP associations, including meta-analysis on summary statistics (*e.g.*, [43, 44]) and fully joint analysis, including Bayesian hierarchical models [45] and mixed models [46]. Extending such methods to haplotype-based analysis in MPPs poses some challenges, including how to best generalize methods to more complex genetic models and for the CC with a limited number of unique genomes. Nonetheless, when multiple levels of molecular traits are measured, joint analyses could conceivably be incorporated into the mediation framework to improve detection power.

## Correlated haplotype effects suggest subtle multi-allelic QTL

Haplotype-based association in MPP allows for the detection of multi-allelic QTL [31], such as potentially observed within the kidney at the local-eQTL of *Pik3c2g*, where mice with B6 contributions in the region have an intermediate level of expression. The correlation coefficient between the haplotype effects for QTL pairs provides an interesting summary, generally not possible in the simpler bi-allelic setting commonly used in variant association analysis. The extent of the correlation between the haplotype effects for QTL pairs of certain genes, such as *Cox7c* (Fig 3A) suggests that these QTL are at least subtly multi-allelic. Correlated haplotype effects are consistent with the genetic regulation, even local to the gene TSS, being potentially complex, likely due to founder-specific modifiers.

## Correlated haplotype effects with differential expression across tissues

We detected numerous eQTL pairs with correlated haplotype effects across tissues but with significantly different magnitudes of overall expression—for example, *Cox7c*, *Ubc*, *Slc44a3*, and *Akr1e1*. We propose two potential explanations for this unique co-occurrence. First, whole tissues differ in their cellular composition. It may be that these genes are primarily expressed in a common cell type whose proportion varies between the different tissue types,

and that the observed differential expression reflects this compositional variation. This hypothesis could be tested by follow-up single cell experiments. Second, each tissue may have additional unique regulatory elements that further modulate expression levels. Uncovering such elements would require in-depth analysis of tissue-specific regulation.

## Mediation analysis

The statistical methods underlying mediation analyses were largely developed in the context of social sciences [3–5], and more recently extended to a genomics setting in which there is generally less experimental control of the relationship between the mediator and outcome. Our mediation analysis approach is adapted from previous studies in DO mice [22, 23, 33] but adds the use of QTL effect sizes to establish consistency with the directionality of the relationships and the formal calculation of an empirical mediation *p*-value through permutation. Related conditional regression approaches were used in the incipient pre-CC lines [47, 48]. Mediation results largely reflect the correlations between the variables after adjusting for additional sources of variation, such as covariates and batch effects. We further require the mediator QTL to have a larger effect size than the outcome QTL in order to identify trios that are consistent with the proposed causal models. It must be emphasized, however, that these steps are not equivalent to experimentally controlling the directionality of the relationship between a gene's expression level and a putative mediator, nor is such control feasible in a large-scale experiment. Variable measurement error on the mediator and gene could flip the perceived directionality of the relationship, resulting in both false positive and negative mediations. Alternatively, the relationships could be more complex than the simple models used here, *e.g.* feedback between the mediator and gene, which these procedures will not detect. Additionally, the causal mediator may not be observed in the data, allowing for other candidates, correlated with the missing causal element, to be incorrectly identified as mediators.

Despite these limitations, the mediation analysis used here provides specific causal candidates for local-eQTL (mediated through chromatin) and distal-eQTL (mediated through nearby genes). For example, we show strong evidence for ZFP985 mediating the genetic regulation of *Akr1e1* (Fig 9). Additional evidence suggests this is done by ZFP985 contributing to reduced *Akr1e1* promoter activity. It is possible that *Zfp985* expression is simply strongly correlated with the true mediator, and others have alternatively proposed *Rex2* expression as a candidate [35], which we did not consider due to low expression in all three tissues. Regardless of the identity of the true mediator, our analysis shows strong evidence that it acts causally by reducing chromatin accessibility near *Akr1e1*.

## QTL mapping power and their effect size estimates

Our reduced sample size of 47 CC strains motivated our use of multiple scopes of statistical significance, from testing for QTL locally to genome-wide. As expected, the additional local-QTL detected by less stringent methods have smaller effect sizes (orange and purple dots for C and L, respectively; S19 Fig). Based on a recent evaluation of QTL mapping power in the CC using simulation [26], this study had approximately 80% power to detect genome-wide QTL with a 55% effect size or greater. Effect size estimates for genome-wide QTL (green dots; S19 Fig) are consistent with this expectation, albeit potentially inflated due to the Beavis effect [49], and provide interpretable point summaries for haplotype-based QTL mapping, analogous to minor allele effect estimates in SNP-based studies. We calculated estimates of effect size in two ways, one based on a fixed effect fitting of the QTL term and the other as a random term [50]. Notably, a small number of the distal-eQTL had low random effect size estimates (S20 Fig) compared with their fixed effects-based estimates, likely the result of outliers with lowly-

observed founder inheritance (*e.g.* rare allele) at the putative QTL. Alternatively, a residual variation estimate, *i.e.* RSS, could be calculated from the shrunken haplotype effects to identify likely false positives, but not be as aggressively reduced as the variance component-based estimates, representing a middle ground approach that was found to be effective [51]. We primarily reported fixed effects estimates due to their consistency with reported expectations [26] for a study of this size.

These QTL mapping results are largely consistent with the molecular traits with detected QTL possessing primarily Mendelian genetic regulation (large effect sizes: > 60%). The relatively limited number of CC strains (< 70 strains) constrains our ability to effectively map QTL for highly complex and polygenic traits. Nevertheless, this study supports the value of CC strains for mapping QTL for simpler traits, such as large effect molecular phenotypes, particularly when considering the further gains that use of replicate observations per strain would yield (not used here). Additionally, joint and/or comparative analyses with the DO and the founder strains can provide strong confirmation of subtle findings in the CC.

## Materials and methods

### Ethics statement

Prior to sacrifice, mice were anesthetized with 100 mg/kg nembutal though intraperitoneal injection. These studies were approved by the Institutional Animal Care and Use Committees (IACUC) at Texas A&M University and the University of North Carolina.

### Animals

Adult male mice (8-12 weeks old) from 47 CC strains were acquired from the University of North Carolina Systems Genetics Core (listed in S1 Appendix) and maintained on an NTP 2000 wafer diet (Zeigler Brothers, Inc., Gardners, PA) and water *ad libitum*. The housing room was maintained on a 12-h light-dark cycle. Our experimental design sought to maximize the number of strains relative to within-strain replications based on the power analysis for QTL mapping in mouse populations [52]; therefore, one mouse was used per strain. Prior to sacrifice, mice were anesthetized with 100 mg/kg nembutal though intraperitoneal injection. Lungs, liver and kidney tissues were collected, flash frozen in liquid nitrogen, and stored at -80°C. These studies were approved by the Institutional Animal Care and Use Committees (IACUC) at Texas A&M University and the University of North Carolina. The experimental design and subsequent analyses performed for this study are diagrammed in Fig 1.

### mRNA sequencing and processing

Total RNA was isolated from flash-frozen tissue samples using a Qiagen miRNeasy Kit (Valencia, CA) according to the manufacturer's protocol. RNA purity and integrity were evaluated using a Thermo Scientific Nanodrop 2000 (Waltham, MA) and an Agilent 2100 Bioanalyzer (Santa Clara, CA), respectively. A minimum RNA integrity value of 7.0 was required for RNA samples to be used for library preparation and sequencing. Libraries for samples with a sufficient RNA integrity value were prepared using the Illumina TruSeq Total RNA Sample Prep Kit (Illumina, Inc., San Diego, USA) with ribosomal depletion. Single-end (50 bp) sequencing was performed (Illumina HiSeq 2500).

Sequencing reads were filtered (sequence quality score $\geq$ 20 for $\geq$ 90% of bases) and adapter contamination was removed (TagDust). Reads were mapped to strain-specific pseudo-genomes (Build37, http://csbio.unc.edu/CCstatus/index.py?run=Pseudo) and psuedo-tran-scriptomes (C57BL/6J RefSeq annotations mapped to pseudo-genomes) using RSEM with

STAR (v2.5.3a). Uniquely aligned reads were used to quantify expression as transcripts per million (TPM) values.

## ATAC-seq processing

Flash frozen tissue samples were pulverized in liquid nitrogen using the BioPulverizer (Biospec) to break open cells and allow even exposure of intact chromatin to Tn5 transposase [53]. Pulverized material was thawed in glycerol containing nuclear isolation buffer to stabilize nuclear structure and then filtered through Miracloth (Calbiochem) to remove large tissue debris. Nuclei were washed and directly used for treatment with Tn5 transposase. Paired-end (50 bp) sequencing was performed (Illumina HiSeq 2500).

Reads were similarly filtered as with RNA-seq. Reads were aligned to the appropriate pseudo-genome using GSNAP (parameter set: -k 15, -m 1, -i 5, –sampling = 1, –trim-mismatch-score = 0, –genome-unk-mismatch = 1, –query-unk-mismatch = 1). Uniquely mapped reads were converted to mm9 (NCBI37) mouse reference genome coordinates using the associated MOD files (UNC) to allow comparison across strains. Reads overlapping regions in the mm9 blacklist (UCSC Genome Browser) were removed. Exact sites of Tn5 transposase insertion were determined as the start position +5 bp for positive strand reads, and the end position -5 bp for negative strand reads [54]. Peaks were called using F-seq with default parameters. A union set of the top 50,000 peaks (ranked by F-seq score) from each sample was derived. Peaks were divided into overlapping 300 bp windows [55]. Per sample read coverage of each window was calculated using coverageBed from BedTools [56].

## Sequence trait filtering for QTL analysis

Trimmed mean of M-values (TMM) normalization (edgeR; [57]) was applied to TPM values from read counts of genes and chromatin windows, respectively. Genes with TMM-normalized TPM values $\leq 1$ and chromatin windows with normalized counts $\leq 5$ for $\geq 50\%$ of samples were excluded (as in [22]) in order to avoid the detection of QTL that result from highly influential non-zero observations when most of the sample have low to no expression. For each gene and chromatin window, we applied $K$-means clustering with $K = 2$ to identify outcomes containing outlier observations that could cause spurious, outlier-driven QTL calls. Any gene or chromatin window where the smaller $K$-means cluster had a cardinality of 1 was removed.

## CC strain genotypes and inferred haplotype mosaics

CC genomes are mosaics of the founder strain haplotypes. The founder haplotype contributions for each CC strain was previously reconstructed by the UNC Systems Genetics Core (http://csbio.unc.edu/CCstatus/index.py?run=FounderProbs) with a Hidden Markov Model [58] on genotype calls [MegaMUGA array [59]] from multiple animals per strain, representing ancestors to the analyzed mice. Notably, QTL mapping power is reduced at loci with segregating variants in these ancestors, and where these specific animals likely differ [60]. To reduce the number of statistical tests, adjacent genomic regions were merged through averaging if the founder mosaics for all mice were similar, defined as L2 distance $\leq 10\%$ of the maximum L2 distance ($\sqrt{2}$ for a probability vector). This reduced the number of tested loci from 77,592 to 14,191.

## Differential expression and accessibility analyses

Read counts for each sample were converted to counts per million (CPM), followed by TMM normalization (edgeR). For chromatin accessibility, windows in which $> 70\%$ of samples had

a CPM $\leq$ 1 were removed, requiring that samples from at least two of the three tissues to have non-zero measurements in order to be considered for differential analysis between tissues. Genes and chromatin windows with no or low counts across sample libraries provide little evidence for detection of differential signal, thus removing them reduces the multiple testing burden. Differentially expressed genes and accessible chromatin windows were determined using *limma* [61], which fit a linear model of the TMM-normalized CPM value as the response and fixed effect covariates of strain, batch, and tissue (lung, liver, or kidney). To account for mean-variance relationships in gene expression and chromatin accessibility data, precision weights were calculated using the *limma* function *voom* and incorporated into the linear modeling procedure. The *p*-values were adjusted using a false discovery rate (FDR) procedure [62], and differentially expressed genes and accessible chromatin windows were called based on the *q*-value $\leq$ 0.01 and log$_2$ fold-change $\geq$ 1. Adjacent significantly differential chromatin windows in the same direction were merged with a *p*-value computed using Simes' method [63], and chromatin regions were re-evaluated for significance using the Simes *p*-values.

## Gene set association analysis

Biological pathways enriched with differentially expressed genes or accessible chromatin were identified with GSAASeqSP [64] with Reactome Pathway Database annotations (July 24, 2015 release). A list of assayed genes were input to GSAASeqSP along with a weight for each gene *g*, calculated as:

$$\text{weight}_g = \text{sign}(\Delta_g) \times (1 - q_g),\tag{1}$$

where $\text{sign}(\Delta_g)$ is the sign of the fold change in gene *g* expression, and $q_g$ is the FDR-adjusted differential expression *p*-value. Pathways with gene sets of cardinality $< 15$ or $> 500$ were excluded.

For pathway analysis of differentially accessible chromatin near genes, each chromatin region was mapped to a gene using GREAT v3.0.0 (*basal plus extension* mode, *5 kb upstream*, *1 kb downstream*, and no distal extension). Weights were calculated as with gene expression, but with $\text{sign}(\Delta_g)$ representing the sign of the fold-change in accessibility of the chromatin region with minimum FDR-adjusted *p*-value that is associated with gene *g*.

## Haplotype-based QTL mapping

QTL analysis was performed for both gene expression and chromatin accessibility using regression on the inferred founder haplotypes [65], a variant of Haley-Knott regression [66, 67] commonly used for mapping in the CC [26, 27, 68–71] and other MPPs, such as *Drosophila* [72].

For a given trait—the expression of a gene or the accessibility of a chromatin region—a genome scan was performed in which at each of the 14,191 loci spanning the genome, we fit the linear model,

$$y_i = \mu + \text{batch}_{b[i]} + \text{QTL}_i + \varepsilon_i\tag{2}$$

where $y_i$ is the trait level for individual *i*, $\mu$ is the intercept, $\text{batch}_b$ is a categorical fixed effect covariate with five levels $b = 1, \ldots, 5$ representing five sequencing batches for both gene expression and chromatin accessibility and where $b[i]$ denotes the batch relevant to *i*, $\varepsilon_i \sim \text{N}(0, \sigma^2)$ is the residual noise, and $\text{QTL}_i$ models the genetic effect at the locus, namely that of the eQTL for expression or the cQTL for chromatin accessibility. Specifically, the QTL term models the (additive) effects of alternate haplotype states and is defined as $\text{QTL}_i = \boldsymbol{\beta}^\text{T}\mathbf{x}_i$, where $\mathbf{x}_i = (x_{i,\text{AJ}}, \ldots, x_{i,\text{WSB}})^\text{T}$ is a vector

of haplotype dosages (*i.e.*, the posterior expected count, from 0 to 2, inferred by the haplotype reconstruction) for the eight founder haplotypes, and $\boldsymbol{\beta} = (\beta_{\text{AJ}}, \ldots, \beta_{\text{WSB}})^{\text{T}}$ is a corresponding vector of fixed effects. Note that in fitting this term as a fixed effects vector, the linear dependency among the dosages in $\mathbf{x}_i$ results in at least one haplotype effect being omitted to achieve identifiability; estimation of effects for all eight founders is performed using a modification described below. Prior to model fitting, to avoid sensitivity to non-normality and strong outliers, the response $\{y_i\}_{i=1}^{n}$ was subject to a rank inverse normal transformation (RINT). The fit of Eq 2 was compared with the fit of the same model omitting the QTL term (the null model) by an F-test, leading to a nominal *p*-value, reported as the logP = $-\log_{10}(p\text{-value})$.

## QTL detection: Local (L), chromosome-wide (C), and genome-wide (G)

QTL detections were declared according to three distinct protocols of varying stringency and emphasis. The first protocol, termed Analysis L, was concerned only with detection of "local QTL", that is, QTL located at or close to the relevant expressed gene or accessible chromatin region. Here we define "local" as ± 10Mb, as been done previously in studies using DO mice [22]. Our intent is to capture QTL that likely act in *cis* on gene expression and chromatin accessibility, which are expected to have strong effects, while also recognizing the limitations of using haplotype blocks with median size of 16.3 Mb [19] and a small sample size. S6 Fig suggest that 10Mb generally captures the strongest QTL signals near the gene TSS and chromatin window midpoint. The second protocol, Analysis C, broadened the search for QTL to anywhere on the chromosome on which the trait is located; the greater number of loci considered meant that the criterion used to call detected QTL for this protocol was more stringent than Analysis L. The third protocol, Analysis G, further broadened the search to the entire genome with the most stringent detection criterion. These protocols used two types of multiple test correction: a permutation-based control of the family-wise error rate (FWER) and the Benjamini-Hochberg False Discovery Rate (FDR; [62]). Below we describe in detail the permutation procedure and then the three protocols.

**Permutation-based error rate control: Chromosome- and genome-wide.** The FWER was controlled based on permutations specific to each trait. The sample index was permuted 1,000 times and recorded, and then genome scans performed for each trait, using the same permutation orderings. Given a trait, the maximum logP from either the entire genome or the chromosome for each permutation was collected and used to fit a null generalized extreme value distribution (GEV) in order to control the genome- and chromosome-wide error rates, respectively [73]. Error rates were controlled by calculating a *p*-value for each QTL based on the respective cumulative density functions from the GEVs: permP = $1 - F_{\text{GEV}}(\text{logP})$, where $F_{\text{GEV}}$ is the cumulative density function of the GEV. We denote genome- and chromosome-wide error rate controlling *p*-values as $\text{permP}_{\text{G}}$ and $\text{permP}_{\text{C}}$ respectively.

The appropriateness of permutation-derived thresholds [74] relies on the CC strains being equally related, thus possessing little population structure and being exchangeable. This assumption was supported by simulations of the funnel breeding design [68]. More recent simulations based on the observed CC strain genomes [26] found non-exchangeable population structure for highly polygenic genetic architectures, albeit at low levels. Nevertheless, we use permutations given that molecular traits often have strong effect QTL that are detectable in the presence of subtle population structure.

**Local analysis—Analysis L.** Our detection criteria for local-QTL leverages our strong prior belief in local genetic regulation. For a given trait, this local analysis involved examining QTL associations at all loci within a 10Mb window of the gene TSS or chromatin region midpoint. A QTL was detected if the $\text{permP}_{\text{C}} < 0.05$. Notably, we can also check whether the

corresponding $permP_G < 0.05$, as an additional characterization of statistical significance of detected local-QTL.

**Chromosome-wide analysis—Analysis C.** Chromosome-wide analysis involved examining QTL associations at all loci on the chromosome harboring the gene or chromatin region in question. The peak logP is input into a chromosome-wide GEV, producing a $permP_C$, which are further subjected to adjustment by FDR to account for multiple testing across the traits [75]. Compared with Analysis L, this procedure is more stringent with respect to local-QTL because of the additional FDR adjustment. In addition, it can detect distal-QTL outside the local region of the trait. Note, since only the most significant QTL is recorded, this procedure will disregard local-QTL if a stronger distal-QTL on the chromosome is observed.

**Genome-wide analysis—Analysis G.** The genome-wide analysis is largely equivalent to Analysis C but examines all genomic loci, while an FDR adjustment is made to the genome-wide $permP_G$. Unlike Analysis C, it incorporates additional scans conditioned on detected QTL to potentially identify multiple QTL per trait (*i.e.* both local- and distal-QTL). Briefly, after a QTL is detected, a subsequent scan (and permutations) is performed in which the previously detected QTL is included in both the null and alternative models, allowing for additional independent QTL to be detected with FDR control. See S2 Appendix for greater detail on this conditional scan procedure and a clear example in S21 Fig. Notably, the local/distal status of the QTL does not factor into Analysis G.

Analyses L, C, and G should detect many of the same QTL, specifically strong local-QTL. Collectively, they allow for efficient detection of QTL with varying degrees of statistical support while strongly leveraging local status.

**QTL effect size.** The effect size of a detected QTL was defined as the $R^2$ attributable to the QTL term; specifically, as $1 - RSS_{QTL}/RSS_0$, where $RSS = \sum_{i=1}^{n} (y_i - \hat{y}_i)^2$ is the residual sum of squares, *i.e.* the sum of squares around predicted value $\hat{y}_i$, and $RSS_{QTL}$ and $RSS_0$ denote the RSS calculated for the QTL and null models respectively. We also calculated a more conservative QTL effect size estimate with a QTL random effects model to compare with the $R^2$ estimate.

**Stable estimates of founder haplotype effects.** The fixed effects QTL model used for mapping, though powerful for detecting associations, is sub-optimal for providing stable estimates of the haplotype effects vector, $\boldsymbol{\beta}$ (calculated as $(\mathbf{X}^T\mathbf{X})^{-1}\mathbf{X}^T\mathbf{y}$, where $\mathbf{X}$ is the full design matrix). This is because, among other things, 1) the matrix of haplotype dosages that forms the design matrix of $\{QTL_i\}_{i=1}^{n}$ in Eq 2, is multi-collinear, which leads to instability, and 2) because the number of observations for some haplotypes will often be few, leading to high estimator variance [24]. More stable estimates were therefore obtained using shrinkage. At detected QTL, the model was refit with $\boldsymbol{\beta}$ modeled as a random effect, $\boldsymbol{\beta} \sim N(\mathbf{0}, \mathbf{I}\tau^2)$ [50], to give an 8-element vector of the best linear unbiased estimates (BLUPs; [76]), $\tilde{\boldsymbol{\beta}} = (\tilde{\beta}_{AJ}, \ldots, \tilde{\beta}_{WSB})^T$. These BLUPs, after being centered and scaled, were then used for further comparison of QTL across tissues.

**Comparing QTL effects across tissues.** To summarize patterns of the genetic regulation of gene expression and chromatin accessibility across tissues, we calculated correlations between the haplotype effects of QTL that map to approximately the same genomic region for the same traits but in different tissues. For cross-tissue pairs of local-QTL, we required that both be detected within the 20Mb window around the gene TSS or the chromatin window midpoint. For cross-tissue pairs of distal-QTL, the QTL positions had to be within 10Mb of each other. All detected QTL were considered, including QTL from Analyses G and C controlled at FDR $\leq 0.2$, allowing for consistent signal across tissues to provide further evidence of putative QTL with only marginal significance in a single tissue.

For a pair of matched QTL $j$ and $k$ from different tissues, we calculated the Pearson correlation coefficient of the BLUP estimates of their haplotype effects, $r_{jk} = \text{cor}(\tilde{\boldsymbol{\beta}}_j, \tilde{\boldsymbol{\beta}}_k)$. Since each $\tilde{\boldsymbol{\beta}}$ is an 8-element vector, the corresponding $r$ are distributed such that $r\sqrt{6}(1 - r^2)^{-1} \sim t_6$ according to the null model of independent variables. Testing alternative models of $r_{jk} > 0$ and $r_{jk} < 0$ produced two $p$-values per pair of QTL. These were then subject to FDR control [62] to give two $q$-values, $q_{ij}^{\{r>0\}}$ and $q_{ij}^{\{r<0\}}$. These were then used to classify cross-tissue pairs of QTL as being significantly correlated or anti-correlated, respectively.

**Variant association.** Variant association has been used previously in MPP (*e.g.*, [51, 77]), and uses the same underlying model as the haplotype-based mapping (Eq 2), with the QTL term now representing imputed dosages of the minor allele. Variant association can more powerfully detect QTL than haplotype mapping if the simpler variant model is closer to the underlying biological mechanism [78], though it will struggle to detect multi-allelic QTL.

Variant genotypes from mm10 (NCBI38) were obtained using the ISVdb [79] for the CC strains, which were converted to mm9 coordinates with the liftOver tool [80]. Variants were filtered out if their minor allele frequencies $\leq 0.1$ or they were not genotyped in one of the CC founder strains to avoid false signals. Tests of association at individual SNPs (variant association) were performed within the local windows of genes with local-QTL detected in more than one tissue. Genes with multiple tissue-selective local-eQTL potentially reflect different functional variants, and could potentially have less consistent patterns of variant association compared with variants that are functionally active in multiple tissues.

## Mediation analysis

Mediation analysis has recently been used with genomic data, including in humans (*e.g.*, [10, 15]) and rodents (*e.g.*, [51, 81]), to identify and refine potential intermediates of causal paths underlying phenotypes. We use a similar genome-wide mediation analysis as used with the DO [22, 33] to detect mediators of eQTL effects on gene expression.

In our study, mediation describes when an eQTL ($X$) appears to act on its target ($Y$) in whole or in part through a third variable ($M$), the mediator, which in this case is a molecular trait. The molecular traits considered here are: a) chromatin accessibility, in which case $X$ serves also as a cQTL (Fig 6A); and, in a separate analysis, b) the expression of a second, distal, gene, in which case $X$ serves also as a distal-eQTL (Fig 6B).

Traditional mediation analysis [3] tests whether the data, for predefined $X$, $Y$ and $M$, are consistent with mediation, doing so in four steps. Steps (1) and (2) establish positive associations $X \rightarrow Y$ and $X \rightarrow M$; this corresponds to our requirement that both the transcript $Y$ and the candidate mediator $M$ have a co-localizing QTL $X$. Step (3) establishes mediation by testing for the conditional association $M \rightarrow Y|X$; this corresponds to testing whether the mediator explains variation in gene expression even after controling for the QTL. Step (4) distinguishes "full mediation", where mediation explains the association between $X$ and $Y$ entirely, *i.e.*, the QTL acts entirely through the mediator, from "partial mediation", where the QTL acts partly through the mediator and partly directly (or through other unmodeled routes).

In our study, we use an empirical approximation of the above adapted to genome-wide data, building on mediation analysis used in studies of DO mice [22, 23, 33]. In outline: For a given eQTL, *i.e.*, a QTL for which step (1) ($X \rightarrow Y$) has been established, we performed a genome-wide mediation scan. This mediation scan consisted of testing step (4) ($X \rightarrow Y|M$), that is, testing whether the eQTL association was significantly reduced when adding the mediator as a covariate, for a large number of "potential" mediators, namely all chromatin regions (or transcripts) genome-wide. Note that most of these potential mediators would be formally ineligible under a traditional analysis (*i.e.*, $X \rightarrow M$ would not hold) but here helped to define a

background (null) level of association. The results of the mediation scan were then filtered to include only results satisfying both of the following criteria: first, the mediator must posess a co-localizing QTL, *i.e.*, a QTL for which step (2) ($X \rightarrow M$) does hold; second, the association between QTL and mediator must be stronger than that between QTL and transcript ($X \rightarrow M > X \rightarrow M$), this approximating step (3) ($M \rightarrow Y|X$). We did not attempt to distinguish between partial and full mediation since this distinction was too easily obscured by noise.

The genome-wide mediation scan procedure in more detail was as follows. Consider an eQTL that our previous genome scan has already shown to affect the expression of gene *j*. Denote the expression level for *j* in individual *i* as $y_i^{G_j}$ and eQTL effect as $\text{eQTL}_i^{G_j}$ (these respectively correspond to $Y$ and $X$ in the mediation description above). Further, consider a proposed mediator $k \in \mathcal{K}$, where $\mathcal{K}$ is the set of all eligible chromatin accessible sites or expressed genes, and let $m_{ik}$ be the value of that mediator for individual *i*. The mediation scan for a given gene/eQTL pair *j* proceeds by performing, for each proposed mediator $k \in \mathcal{K}$, a model comparison between the alternative model,

$$y_i^{G_j} = \mu + \text{eQTL}_i^{G_j} + m_{ik} + \text{batch}_{b[i]} + \varepsilon_i, \tag{3}$$

and the null model,

$$y_i^{G_j} = \mu + m_{ik} + \text{batch}_{b[i]} + \varepsilon_i,$$

where other terms are as described for Eq 2. (Not, shown are additional covariates, such as conditioned loci, which would be included in both the null and alternative models.) The above model comparison can be seen as re-evaluating the significance of a given eQTL association by conditioning on each proposed mediator in turn (*i.e.*, testing $X \rightarrow Y|M$ for each $M$). The resulting mediation scan, in contrast to the earlier-described genome scans, thus fixes the QTL location while testing across the genome for candidate mediators.

In general, assuming most proposed mediators are null, the mediated logP should fluctuate around the original eQTL logP, since the model comparison will resemble the original test of that locus in the genome scan. For mediators that possess some or all of the information present in the eQTL, however, the mediated logP will drop relative to the original logP, reflecting $X \rightarrow Y|M$ being less significant than $X \rightarrow Y$. Empirical significance of genome-wide mediation has previously been determined by comparing the nominal mediator scores to the distribution of mediation scores genome-wide, the latter effectively acting as a null distribution [22, 23, 33]. We instead determine a null distribution explicitly by permutation, characterizing the distribution of the minimum logP (as opposed to maximum logP for QTL scans) to estimate significance and set false positive control (FWER). As with the QTL mapping procedures, we performed mediation scans on 1,000 permutations, permuting the mediators rather than the outcome, to characterize GEVs from the minimum logP, that is, $\text{permP}^m = F_{\text{GEV}}(\text{logP})$, at both chromosome- and genome-wide levels.

Detection of mediation is dependent on a number of assumptions about the underlying variables, their relationships, and those relationships' directionality [4]. Many of these cannot be controlled in a system as complex as chromatin accessibility and transcriptional regulation in whole living organisms. Nevertheless, signals in the data that are consistent with the mediation model represent candidate causal factors that regulate gene expression. Though it is impossible to ensure that the direction of causation is indeed $M \rightarrow Y$, as assumed by the mediation analysis, we require additional checks to formally declare mediation of an eQTL. The presence of the relationship $X \rightarrow Y$ is already established in that mediation scans are only performed for detected eQTL. In addition for any candidate mediator with a significant $\text{permP}^m$, we require detection of a mediator QTL: $X \rightarrow M$. These requirements, as mentioned earlier,

are consistent with traditional mediation analysis. Finally, in an attempt to identify mediator-to-gene relationships consistent with the proposed models, we require that $X \rightarrow M$ is more significant than $X \rightarrow Y$. Though this step cannot confirm the directionality of the relationship between $M$ and $Y$, as variable noise level between $M$ and $Y$ could reduce the estimated effect size of their respective QTL, it will identify candidate mediators that are consistent with the proposed models. Further details on the mediation analysis are described in S3 Appendix, including the permutation approach and the formal criterion by which mediation is declared for both chromatin and gene mediators (Fig 6).

## Software

All statistical analyses were conducted with the R statistical programming language [82]. The R package miQTL was used for all the mapping and mediation analyses, and is available on GitHub at https://github.com/gkeele/miqtl and a fixed version as S8 File.

## Supporting information

**S1 Fig. Principle components analysis identifies tissue type as key source of variation for gene expression and chromatin accessibility.** Molecular traits for liver (purple), lung (green), and kidney (orange) tissue samples were derived from RNA-seq and ATAC-seq data. Principal components (PC) 1 and 2 capture a majority of the variation and show a greater amount of between tissue variability than within tissue variability.
(TIF)

**S2 Fig. Concordance between differentially expressed genes and differentially accessible regions in between-tissue comparisons.** Genes were categorized by the direction of the difference in expression and chromatin accessibility in their promoter regions.
(TIF)

**S3 Fig. Overlap across tissues of (A) genes and (B) chromatin windows used for QTL analysis.** Sequence traits were filtered to remove outcomes more likely to cause spurious QTL signals. Genes with TPM $\leq 1$ and chromatin windows with TMP $\leq 5$ for $\geq 50\%$ of samples were removed from analysis. After this filtering process, lung had the greatest number of traits analyzed, for both genes and chromatin windows, followed by kidney and then liver.
(TIF)

**S4 Fig. Overlap across tissues of (A) genes and (B) chromatin windows with local-QTL detected.** The majority of sequence traits with a local-QTL detected were identified in only a single tissue. Kidney had the highest number of local-eQTL, whereas lung had the highest number of local-cQTL. Liver had a relative lack of local-cQTL, which may relate to its having the fewest chromatin windows analyzed (S3B Fig). Results included local-QTL detected with Analysis G (FDR $\leq 0.1$), Analysis C (FDR $\leq 0.1$), and Analysis L (genome-wide and chromosome-wide).
(TIF)

**S5 Fig. QTL mapping results using only Analysis G or Analysis C.** QTL map plots of (A) eQTL and (B) cQTL with FDR controlled at 0.1 and 0.2 for liver, lung, and kidney. Detected QTL from Analysis G (multi-stage FDR) and Analysis C (chromosome-wide FDR) are included. Analysis C, which uses FDR control for chromosome-wide significant QTL, produces a large number of intra-chromosomal distal-QTL. The y-axis represents the genomic

position of the gene or chromatin site, and the x-axis represents the genomic position of the
QTL. Local-QTL appear as dots along the diagonal.
(TIF)

**S6 Fig. Highly significant QTL map nearby the gene TSS and chromatin window midpoint.**
The permutation-based *p*-value (permP) from (A) Analysis G and (B) Analysis C for eQTL
and cQTL by their distance (Mb) from the gene TSS and the midpoint of the chromatin site.
Inter-chromosomal distal-QTL are not included. The red dashed lines represent ±10Mb of the
gene TSS or the midpoint of the chromatin site for classifying QTL as local or distal. Signifi-
cant signals (yellow or blue), based on FDR ≤ 0.1, are largely local. Analysis C detects many
more intra-chromosomal distal-QTL.
(TIF)

**S7 Fig. QTL effect size by local/distal status.** Each dot represents a QTL detected through
either (A) Analysis G or (B) Analysis C with FDR ≤ 0.1. The three horizontal bars represent
the 25th, 50th, and 75th quantiles of QTL effect sizes for all local-QTL per tissue. More local-
eQTL are detected and have higher effects than distal-QTL. Analysis C detects a large number
of intra-chromosomal distal-QTL that Analysis G does not, many of which have low effect
sizes. Effect size estimates are based on a fixed effects model.
(TIF)

**S8 Fig. CAST and PWK haploytpes have more extreme effects for (A) eQTL and (B) cQTL
compared with the other strains.** Haplotype effects were estimated as BLUPs, which are con-
strained and centered around 0. Each QTL is represented by an 8-element effect vector. Foun-
ders with more extreme effects are identified by comparing the absolute values of effects.
Founder haploytpe effect trends for eQTL are similar to cQTL. The trends are unstable in dis-
tal-cQTL because so few are identified.
(TIF)

**S9 Fig. Effect sizes between cross-tissue QTL pairs are lowly but significantly correlated.**
Comparisons of QTL effects sizes between (liver/lung) are in the left column, (liver/kidney)
middle column, and (lung/kidney) right column. eQTL are yellow and cQTL are blue. Local-
eQTL are plotted in the top row, distal-eQTL in the second row, local-cQTL in the third row,
and distal-cQTL in the bottom row, with only four pairs detected in (lung/kidney).
(TIF)

**S10 Fig. Consistent genetic regulation of gene expression and chromatin accessibility
observed across tissues.** There was an excess of significant positively correlated haplotype
effects in QTL pairs across tissues for gene expression and chromatin accessibility. Pairs of
QTL observed in multiple tissues were defined for local-eQTL (left column), distal-eQTL
(middle column), and local-cQTL (right column). Only four pairs of distal-cQTL were
observed, all shared between lung and kidney. A right-tailed test the correlation between hap-
lotype effects ($H_A: r > 0$) was performed for each QTL pair, producing *p*-values that were then
FDR adjusted. Null simulations of uncorrelated 8-element vector pairs for each class of QTL
and pairwise tissue comparison emphasize the observed enrichment in correlated haplotype
effects between QTL pairs.
(TIF)

**S11 Fig. Cross-tissue QTL pairs with highly correlated haplotype effects map proximally
to each other.** Haplotype effects were estimated as constrained BLUPs. Pairwise correlations
of the 8-element effect vectors were calculated for QTL pairs, and plotted again the distance
between the QTL coordinates in Mb for (liver/lung) in the left column, (liver/kidney) in the

middle column, and (lung/kidney) in the right column. Single eQTL and cQTL pairs are represented as a yellow and blue dots, respectively. Local-eQTL are shown in the top row, distal-eQTL in the second row, local-cQTL in the third row, and distal c-QTL in the bottom row, for which only four pairs were detected in (lung/kidney).
(TIF)

**S12 Fig. The gene *Ubc* has consistent strong local-eQTL observed in the three tissues.** The local-eQTL consistently drove higher expression when the B6, NOD, NZO, and WSB haploytpes were present. Expression levels in liver and lung were found to be significantly different ($q = 0.022$). The estimated haplotype effects were highly consistent with the expression data, represented as interquartile bars categorized by most likely diplotype. The haplotype and variant associations in the eQTL regions were similar across tissues, suggesting they may represent the same causal origin. The red tick represents the *Ubc* TSS, the black tick represents the peak variant association, and the colored ticks represent the peak haplotype association for each tissue.
(TIF)

**S13 Fig. The gene *Rnf13* has unique patterns of genetic regulation across tissues.** A strong local-eQTL was detected in liver, and after conditioning on it, a statistically significant distal-eQTL was detected (Analysis G) on chromosome 12, largely driven by the B6 haplotype, distinct from the local-eQTL. The unique haplotype effect patterns for each eQTL can be seen in both the expression data, represented by interquartile bars for most likely diplotype, and the estimated effects. The red tick marks the *Rnf13* TSS and the black tick marks the location of distal-eQTL. Another strong distal-eQTL was detected on the X chromosome in lung.
(TIF)

**S14 Fig. Co-localizing eQTL and cQTL are not sufficient for statistical mediation.** The approach used to detect mediation through chromatin accessibility requires that the eQTL and cQTL co-localize (both within 10Mb of the gene TSS), as well as possess similar haplotype effect patterns. Co-localizing cQTL are observed for local-eQTL for both (A) *Hdhd3* in liver and (B) *Acbd4* in kidney. QTL and mediation scans are shown, with chromosomes 4 and 11 blown up for *Hdhd3* and *Acbd4*, respectively. The red ticks denote the TSS for both genes. The haplotype effects for the eQTL and cQTL are highly correlated ($r = 0.96$) for *Hdhd3*, but not for *Acbd4* ($r = 0.55$). Strong mediation of the *Hdhd3* eQTL through chromatin is detected, but not for *Acbd4*. The effect size of the co-localizing cQTL to *Acbd4* is smaller than its eQTL, also inconsistent with the relationship depicted in Fig 6A.
(TIF)

**S15 Fig. Mediation of *Ccnyl1* distal-eQTL through *Zfp979* expression.** Expression of *Ccnyl1* and *Zfp979* are correlated ($r = 0.72$) in lung, which is also observed in the expression data categorized by diplotype and the haplotype effects. The distal-eQTL on chromosome 4 for *Ccnyl1* corresponds closely to local-eQTL of *Zfp979*. *Ccnyl1* is located on chromosome 1, indicated by the red tick. *Zfp979* and *Zfp985*, both zinc finger proteins likely with DNA binding properties, are identified as strong candidate mediators of the distal-eQTL at genome-wide significance. The correlations, magnitude of effects, and mediation are consistent with the simple relationship depicted in the graph. The distal-eQTL and candidate mediators are located in a region of interest that regulates *Akr1e1* expression.
(TIF)

**S16 Fig. Mediation of *Akr1e1* distal-eQTL through *Zfp985* expression.** Expression of *Akr1e1* and *Zfp985* are anti-correlated ($r = -0.69$) in lung. This relationship is also observed in the

expression data with bars representing the interquartile range, categorized by most likely diplotype, and the haplotype effects. The QTL and mediation scans reveal that *Akr1e1*, with TSS marked with a red tick on chromosome 13, possesses a distal-eQTL on chromosome 4 that is nearby the strong local-eQTL of *Zfp985*. The mediation scan identifies *Zfp985* as a strong candidate mediator consistent with the mediation model. A more complete picture of the genetic regulation of *Akr1e1* expression is pieced together by looking across all three tissues and includes a potential chromatin mediator (Fig 9).
(TIF)

**S17 Fig. Confirmation of *Akr1e1* distal-eQTL and mediation by *Zfp985* in kidney tissue of Diversity Outbred mice.** A genome-wide significant distal-eQTL was detected for *Akr1e1* in liver, lung (shown here), and kidney tissues from 47 CC strains. In a larger sample of kidney tissue from outbred DO mice, the same distal-eQTL and mediation relationship were observed. As expected, the larger sample of the DO results in greater statistical significance, and confirms that the NOD effect is more strongly negative than NZO, PWK, and WSB, which the haplotype effects plots for the *Zfp985* local-eQTL suggested. Notably, *Zfp985* was not tested in the CC kidney because of low expression levels, though the distal-eQTL for *Akr1e1* is consistent with its activity, which is here confirmed in the DO.
(TIF)

**S18 Fig. Observed relationships across the three tissues related to the genetic regulation of *Akr1e1* expression.** The model for the distal genetic regulation of *Akr1e1* expression, described in Fig 9, was reconstructed from these observed relationships. Solid arrows were observed, whereas dashed arrows are assumed. QTL effect sizes represent the proportion of variance explained by the QTL and mediation *p*-values (permP) were defined using a permutation procedure. The assumed relationships are supported by the presence of the distal-eQTL in all three tissues. The *Zfp985* mediator relationship in kidney, though not observed in the CC, was observed in the related DO population.
(TIF)

**S19 Fig. Local-QTL effect sizes by mapping analysis.** Based on ranking mapping analyses with respect to the extent of scope, local (L; magenta) to chromosome (C; plum) to genome-wide (G; cyan), the greater the scope corresponded to reduced power to detect QTL, shown in liver, lung, and kidney tissues for gene expression (yellow line) and chromatin accessibility (blue line). Each dot represents a detected local-QTL, colored according to the highest scope mapping procedure that detected it. The three horizontal bars represent the 25th, 50th, and 75th quantiles of QTL effect sizes for all local-QTL per tissue. Analysis G generally detects QTL with effect size > 60%, whereas Analyses C and L detect QTL effect sizes > 45%. Effect size estimates correspond to a fixed effects model of the QTL.
(TIF)

**S20 Fig. Comparison of QTL effect sizes estimates from fixed effects and random effects models.** The effect size from the random effect fit is harshly penalized compared with the fixed effect estimate, likely due to a sample size of 47 mice. Notably, there are a number of distal-eQTL that are more harshly reduced by the random effects model compared with the other QTL, likely representing signals resulting from extreme observations or imbalances in founder contributions at the locus. QTL detected by Analysis G (FDR ≤ 0.1), C (FDR ≤ 0.1), and L are shown.
(TIF)

**S21 Fig. Detection of local-eQTL after conditioning on distal-eQTL for *Gpn3*.** The multi-stage conditional regression approach of Analysis G allows for the detection of multiple

genome-wide significant QTL, which can be appropriately incorporated into an FDR proce-
dure across many outcomes. In this example in lung tissue, the gene *Gpn3* initially has a strong
distal-eQTL on chromosome 8 [top left]. Though a peak is detected near the TSS of *Gpn3*, it
does not meet genome-wide significance. However, after conditioning on the distal-eQTL, the
local-eQTL is detected [bottom left]. Horizontal dashed lines represent empirical 95% signifi-
cance thresholds based on 1,000 permutations.
(TIF)

**S1 Appendix. CC strains used in study.**
(PDF)

**S2 Appendix. Detailed description of conditional genome-wide scans (Analysis G).**
(PDF)

**S3 Appendix. Detailed description of mediation analysis.**
(PDF)

**S1 Table. Number of differentially expressed genes and accessible chromatin regions
detected between liver, lung, and kidney tissues.**
(PDF)

**S2 Table. Number of genes with eQTL detected in liver, lung, and kidney tissues at
FDR $\leq$ 0.1.**
(PDF)

**S3 Table. Number of genes with eQTL detected in liver, lung, and kidney tissues at
FDR $\leq$ 0.2.**
(PDF)

**S4 Table. Number of chromatin accessibility sites with cQTL detected in liver, lung, and
kidney tissues at FDR $\leq$ 0.1.**
(PDF)

**S5 Table. Number of chromatin accessibility sites with cQTL detected in liver, lung, and
kidney tissues at FDR $\leq$ 0.2.**
(PDF)

**S6 Table. Number of genes with chromatin mediation of local-eQTL in liver, lung, and kid-
ney tissues.**
(PDF)

**S7 Table. Genes with distal-eQTL with gene mediators detected in lung and kidney tissues.**
(PDF)

**S1 File. Differentially expressed genes across comparisons of three tissues.**
(CSV)

**S2 File. Differentially accessible chromatin regions across comparisons of three tissues.**
(CSV)

**S3 File. Local-eQTL detected across three tissues.**
(CSV)

**S4 File. Distal-eQTL detected across three tissues.**
(CSV)

**S5 File. Local-cQTL detected across three tissues.**
(CSV)

**S6 File. Distal-cQTL detected across three tissues.**
(CSV)

**S7 File. Mediation of local-eQTL through chromatin state detected across three tissues.**
(CSV)

**S8 File. A frozen version (1.1.2) of miQTL, the R package used to perform QTL and mediation analyses.**
(ZIP)

**S9 File. R code to generate all figures in the manuscript.**
(R)

**S10 File. R code to generate all supplemental figures and tables.**
(R)

**S11 File. R code to generate plots used in the manuscript and supplemental figures.**
(R)

**S12 File. R code for repeated analyses used in manuscript.**
(R)

**S13 File. R code to pull QTL and mediation numbers for tables.**
(R)

**S14 File. R code tutorial for using miQTL to run QTL and mediation analyses reported in the manuscript.**
(R)

## Acknowledgments

We thank Lucas T. Laudermilk for his insights on the functional mechanisms of zinc finger protein-induced gene silencing, and Samir N. P. Kelada and Lauren J. Donoghue for reading the manuscript and providing thoughtful suggestions, all from UNC. We thank Fred A. Wright of North Carolina State University for helpful discussions on eQTL and mediation analyses. We thank Gary A. Churchill of The Jackson Laboratory for discussions on mediation analysis that influenced this work.

## Author Contributions

**Conceptualization:** Gregory R. Keele, Gregory E. Crawford, William Valdar, Ivan Rusyn, Terrence S. Furey.

**Data curation:** Jennifer W. Israel, Jeremy M. Simon, Paul Cotney.

**Formal analysis:** Gregory R. Keele, Bryan C. Quach.

**Funding acquisition:** Gregory E. Crawford, Ivan Rusyn, Terrence S. Furey.

**Investigation:** Grace A. Chappell, Lauren Lewis, Alexias Safi.

**Methodology:** Gregory R. Keele, William Valdar.

**Project administration:** Gregory E. Crawford, William Valdar, Ivan Rusyn, Terrence S. Furey.

**Software:** Gregory R. Keele, Bryan C. Quach.

**Supervision:** Gregory E. Crawford, William Valdar, Ivan Rusyn, Terrence S. Furey.

**Visualization:** Gregory R. Keele, Bryan C. Quach.

**Writing – original draft:** Gregory R. Keele, Bryan C. Quach.

**Writing – review & editing:** William Valdar, Ivan Rusyn, Terrence S. Furey.

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
