## [Decision Letter · Decision Letter 0]

26 Aug 2019

Dear Dr Keele,

Thank you very much for submitting your Research Article entitled 'Integrative QTL analysis of gene expression and chromatin accessibility identifies multi-tissue patterns of genetic regulation' to PLOS Genetics.

The manuscript was fully evaluated at the editorial level and by three peer reviewers. As you will see, the reviewers appreciated the attention to an important problem, but raised a number of concerns about the current manuscript. Based on the reviews, we will not be able to accept this version of the manuscript, but we would be interested to review again a much-revised version. We cannot, of course, promise publication at that time.

If you decide to revise the manuscript for further consideration at PLOS Genetics, please aim to resubmit within the next 60 days, unless it will take extra time to address the concerns of the reviewers, in which case we would appreciate an expected resubmission date by email to plosgenetics@plos.org.

[LINK]

We are sorry that we cannot be more positive about your manuscript at this stage. Please do not hesitate to contact us if you have any concerns or questions.

Yours sincerely,

Gregory S. Barsh

Editor-in-Chief

PLOS Genetics

Gregory Copenhaver

Editor-in-Chief

PLOS Genetics

Reviewer's Responses to Questions

**Comments to the Authors:**

Reviewer #1: Uploaded as attachment "plosG_reviews"

Reviewer #2: The authors analyzed RNA-seq and ATAC-seq data from three tissues of 47 strains of the Collaborative Cross (CC) mouse population. They did a series of analyses to study genetic regulation, including differential analysis, pathway enrichment analysis, QTL analysis, and mediation analysis. Below I provide some comments on the mediation analysis.

Major comments:

1. Page 18, Mediation analysis, “the eQTL is always included in the alternative model but not the null model.” I am not sure why not instead exclude the mediator in the null model. It seems that excluding the eQTL in the null model is to test partial mediation vs. full mediation. Based on my understanding, mediation analysis is to test the red mediation path in Fig 6 [Refs 9, 56], not the yellow eQTL path.

Then in S3 Appendix, “Instead of permuting the trait outcome, the pairing of trait to QTL is maintained, and the mediator is permuted.” It seems the mediator was tested. It is confusing, and clarification should be given.

2. The permutation-based p-values were calculated using a nonstandard approach based on generalized extreme value distribution (GEV). Is this method designed for genetically correlated samples? Is it more efficient than simply comparing the observed statistic with the null statistics? Is there a way to empirically prove that this approach works for the analyzed data?

In S3 Appendix, “An FDR adjustment is not used with mediation because it is only evaluated for detected QTL, which violates the uniform null distribution expectations of FDR.” At least comparing the observed statistic with the null statistics can control FDR, which is more liberal than family-wise error rate (FWER) control. It can be more powerful, especially when the sample size is only 47, and the testing is genome-wide.

3. The R package miQTL seems to lack helpfile/tutorial for functions related to mediation analysis. Those would make the scheme of mediation analysis clearer. Or in the paper, providing a concise algorithm to list each step of the mediation analysis would help. Separating gene-gene and chromatin-gene mediation analyses would help further. Right now, the methods of mediation analysis are not described clearly.

Minor comments:

Fig 1, “ATAC-seq” was misspelled as “ATATC-seq.”

Line 17, Ref 9 seems mis-cited. It is not related to “ribosome occupancy with protein abundances.”

Fig S1, caption, “liver (pink).” The pink color seems to be purple from non-color-blind eyes.

The captions for the supplementary figures shown in the main text file and the supplementary file do not match.

Reviewer #3: This paper describes a resource of eQTLs and chromatin-QTLs in mice. The authors performed RNA-seq and ATAC-seq in 3 tissues in 47 mouse strains from the Collaborative Cross. The authors analyze tissue-specific and shared effects, and show examples of colocalizing eQTL and chromatin-QTL effects that suggest regulatory mechanisms in cis and trans.

Overall, the results of the study are in line with findings from previous QTL studies, and highlight new examples of genetic associations in the mouse.

Comments:

• Mediation analysis:

The authors use mediation analysis to identify eQTL and chromatin-QTL signals that may result from the same causal variant. The authors suggest that a significant result in the mediation analysis implies that the chromatin peak regulates the gene. In fact, there are other explanations for this finding (reverse causality, pleiotropy, multiple linked causal variants). The authors refer to such issues in the Methods (Page 18, line 656). This caveat should be discussed explicitly in the main text.

The issue of the potential for multiple linked causal variants deserves particular attention in the Collaborative Cross, given the potential for large haplotype blocks with many tightly linked variants. It would be helpful to remind readers of the approximate haplotype size / rate of recombination in the Collaborative Cross, which determines the resolution at which it is possible to pinpoint causal genetic variants in this study. Then, this should be discussed and analyzed in more detail with respect to colocalizing eQTL and chromatin-QTLs. For any given pair of colocalizing eQTL and chromatin-QTLs, how many variants are in perfect LD? This affects our interpretation of how often the chromatin peak indeed causally mediates the effect on gene expression, versus how often the two effects might be caused by different tightly linked variants.

The initial step in the pairing procedure involves looking for signals located within 10 Mb of each other. However, the fact that there are eQTLs for the same gene (or cQTLs for the same peak) does not indicate that such QTLs could be caused by the same variant, and the authors follow up with mediation analysis. It would improve clarity to omit the results on page 5, lines 121-127; and instead only report the numbers in lines 130-133.

Additional evidence that their statistical approach for detecting mediation is well-calibrated (e.g., qq plots, power analysis, etc. based on real data and simulated data) would strengthen the paper.

• Power analysis:

The value of this resource would be enhanced if the results of each variant-gene-tissue QTL test were reported (regardless of significance). Furthermore, the authors should calculate and report the power to detect effects of a given size for each variant-gene-tissue test (e.g., What is the power to detect 25% or 50% effect sizes? What is the smallest effect size that they are 80% powered to detect?). The authors report that, on average, the study has approximately 80% power to detect 55% effect sizes — but, with the data in hand, this can now be precisely calculated for every QTL test to account for the expression of the specific gene, etc. This would allow future users of this resource to understand whether, in cases where no eQTL is detected, what effect sizes the study was powered to detect.

Information on power is furthermore very important for assessing whether a QTL is “tissue-specific”. If the test for a given variant-gene pair has substantially different power in two different tissues, this could lead to it being detected in one tissue but not the other for technical reasons. Was power considered in assessing tissue specificity?

• “These QTL mapping results are largely consistent with primarily Mendelian genetic regulation of molecular traits.” This implication does not seem correct to me, as necessarily the analysis is biased towards finding large effects. The study is not powered to find small-effect, polygenic regulation of molecular traits. The results here do not broadly inform the genetic architecture of molecular traits, other than that, for some genes, large-effect associations do exist.

Minor points:

• Fig. S21. Panels (A) and (B) are referred to in the caption but not labeled in the plot.

• Page 12, Line 365: “our analysis has shown strong evidence that it acts causally by inducing a heterochromatic state near the Akr1e1 promoter”. Accessible/non-accessible does not strictly correspond to euchromatin/heterochromatin. Furthermore, the evidence presented does not suggest, as the authors seem to be implying, that ZFP985 acts directly on the promoter of Akr1e1. I would suggest: “… that it leads to reduced accessibility at the Akr1e1 promoter”

**Have all data underlying the figures and results presented in the manuscript been provided?**

Reviewer #1: Yes

Reviewer #2: None

Reviewer #3: No: Raw genotype and RNA/ATAC data are not yet available. Full summary statistics of association tests and mediation analysis are not available.

PLOS authors have the option to publish the peer review history of their article (what does this mean?). If published, this will include your full peer review and any attached files.

Reviewer #1: No

Reviewer #2: No

Reviewer #3: No

---

## [Decision Letter · Decision Letter 1]

23 Nov 2019

Dear Dr Keele,

We are pleased to inform you that your manuscript entitled "Integrative QTL analysis of gene expression and chromatin accessibility identifies multi-tissue patterns of genetic regulation" has been editorially accepted for publication in PLOS Genetics. Congratulations!

Yours sincerely,

Gregory S. Barsh

Editor-in-Chief

PLOS Genetics

Gregory Copenhaver

Editor-in-Chief

PLOS Genetics

Comments from the reviewers (if applicable):

Reviewer's Responses to Questions

**Comments to the Authors:**

Reviewer #1: The authors have addressed to a satisfactory level most of the comments raised.

Reviewer #2: My comments have been addressed.

Reviewer #3: The authors have addressed my comments.

**Have all data underlying the figures and results presented in the manuscript been provided?**

Reviewer #1: Yes

Reviewer #2: None

Reviewer #3: Yes

PLOS authors have the option to publish the peer review history of their article (what does this mean?). If published, this will include your full peer review and any attached files.

Reviewer #1: No

Reviewer #2: No

Reviewer #3: No

**Data Deposition**

http://datadryad.org/submit?journalID=pgenetics&manu=PGENETICS-D-19-00860R1

**Press Queries**

---

## [Editor Report · Acceptance letter]

8 Jan 2020

PGENETICS-D-19-00860R1 

Integrative QTL analysis of gene expression and chromatin accessibility identifies multi-tissue patterns of genetic regulation 

Dear Dr Keele, 

We are pleased to inform you that your manuscript entitled "Integrative QTL analysis of gene expression and chromatin accessibility identifies multi-tissue patterns of genetic regulation" has been formally accepted for publication in PLOS Genetics! Your manuscript is now with our production department and you will be notified of the publication date in due course.

With kind regards,

Matt Lyles

PLOS Genetics

On behalf of:
